



Geographic variation and temporal trends in ice phenology in Norwegian lakes over a century
Jan Henning L'Abée-Lund[1], Leif Asbjørn Vøllestad[2], John Edward Brittain[1,3], Ånund Sigurd Kvambekk[1]
and Tord Solvang[1]
[1] Norwegian Water Resources and Energy Directorate, Box 5091 Majorstuen, N-0301 Oslo, Norway
[2] Centre for Ecological and Evolutionary Synthesis, University of Oslo, Box 1066 Blindern, N-0316
Oslo, Norway
[3] Natural History Museum, University of Oslo, Box 1072 Blindern, N-0316 Oslo, Norway
*Correspondence to*: Jan Henning L'Abée-Lund (jlabeelund@gmail.com)



Abstract
Long-term observations of ice phenology in lakes are ideal for studying climatic variation in time and
space. We used a large set of observations from 1890 to 2020 of the timing of freeze-up and break-
up, and the length of ice-free season, for 101 Norwegian lakes to elucidate variation in ice phenology
across time and space. The dataset of Norwegian lakes is unusual, covering considerable variation in
altitude (4 – 1401 m a.s.l.) and climate (from oceanic to continental) within a substantial latitudinal
and longitudinal gradient (58.2 – 69.9 °N; 4.9 – 30.2 °E).
The average date of ice break-up occurred later in spring with increasing altitude, latitude and
longitude. The average date of freeze-up and the length of the ice-free period decreased significantly
with altitude and longitude. No correlation with distance from the ocean was detected, although the
geographical gradients were related to regional climate due to adiabatic processes (altitude), solar
radian (latitude) and the degree of continentality (longitude).  There was a significant lake area effect
as small lakes froze-up earlier due to less volume. There was also a significant trend that lakes were
completely frozen over later in the autumn in recent years. After accounting for the effect of long-
term trends in the large-scale NAO index, a significant but weak trend over time for earlier ice break-
up was detected.
An analysis of different time periods revealed significant and accelerating trends for earlier break-up,
later freeze-up and completely frozen lakes after 1991. Moreover, the trend for a longer ice-free
period also accelerated during this period, although not significant.
An understanding of the relationship between ice phenology and geographical parameters is a
prerequisite for predicting the potential future consequences of climate change on ice phenology.
Changes in ice phenology will have consequences for the behaviour and life cycle dynamics of the
aquatic biota.
Keywords: Lake ice, Ice phenology, Climate change, Lake characteristics, Geographical variation



## 1 Introduction

Lakes make up a substantial part (15-40 %) of the arctic and sub-arctic regions of the Northern Hemisphere (Brown and Duguay 2010). Most of these lakes freeze over annually. In addition to its substantial biological importance (Prowse 2001), this annual freezing has significant repercussions for transportation, local cultural identity and religion (Magnusson et al., 2000; Sharma et al., 2016; Knoll et al., 2019). The importance of freshwater and ice formation for people has resulted in the monitoring of freezing and thawing of lake ice for centuries (Sharma et al., 2016).

Lakes and their ice phenology are effective sentinels of climate change (Adrian et al., 2009) and ice phenology has been studied extensively (e.g., reviewed by Brown and Duguay, 2010). In general, freeze-up occurs later and break-up appears earlier on global (Magnuson et al., 2000; Benson et al., 2012; Du et al., 2017), regional (Duguay et al., 2006; Mishra et al., 2011; Hewitt et al., 2018) and local scales (Choiński et al., 2015; Takács et al., 2018). Despite these general results, the strength of the trends varies among studies. The time of freeze-up was delayed by 0.3 to 5.7 days/decade (Benson et al. 2012, Magnusson et al. 2000), whereas the timing of ice break-up was delayed by between 0.2 and 6.3 days/decade (Mishra et al., 2011; Magnusson et al., 2000). Some of this variation is a consequence of differences in the length of the study period, covering from more than a century to just a single decade. This wide variation in time period and the particular time-period studied is important to consider when trying to compare the strength of trends in ice phenology parameters. Global mean temperature has changed considerably after 1880 (Hansen et al., 2006), and the change (increase) in temperature is particularly evident in later decades. By dividing data from the 1931-2005 period into shorter timer periods, Newton and Mullan (2020) showed, for Fennoscandia, an increase in the magnitude of the general trend in earlier break-up in 1991-2005 compared to earlier periods. In North America the trend was for earlier break-up, but it was neither spatially nor temporally consistently explained by local or regional variation in climate (Jensen et al., 2007).

In Fennoscandia, recording ice phenology has long traditions due to the importance of frozen lakes and rivers for transport and recreation (Sharma et al., 2016). Data from Swedish and Finnish lakes have been studied in detail by Eklund (1999), Blenckner et al. (2004) and Palecki & Barry (1986). Based on Swedish data for the period 1710-2000, Eklund (1999) showed that ice break-up did not change from 1739 to 1909, became 5 days earlier in the period 1910-1988 and still 13 days earlier during the final period (1988-1999). Furthermore, ice freeze-up was later in the 1931-1999 period than in the 1901-1930 period. Similarly, stronger trends in both freeze-up and break-up in the last decade of the 1950-2009 time period have been shown for both Finnish and Karelian lakes (Blenckner et al., 2004; Efremova et al., 2004). Moreover, Blenckner et al. (2004) showed that large



variability was apparent south of 62° N, indicating that lakes in southern Sweden were more
influenced by large-scale climate effects (such as the North Atlantic Oscillation; NAO (Hurrell, 1995))
than northern lakes. This pattern was explained by the mountain range between Norway and
Sweden affecting the regional circulation in the north.
Despite the fact that registration of ice phenology has been undertaken in a large number of lakes
and rivers in Norway, as early as 1818 in some lakes (www.nve.no), few lakes have been studied in
detail and no country-wide analysis has been done. Trends in freeze-up and break-up have been
analyzed for two subalpine lakes in Central Norway (Kvambekk and Melvold, 2010; Tvede, 2004).
Although not covering the exact same period, both freeze-up and break-up show different trends in
the two lakes. Although geographically close to lakes in Sweden and Finland, Norwegian lakes
demonstrate considerably more variation in topography and climate. Norwegian lakes, situated in
the western parts of the Scandinavian peninsula, encompass a large a variation in altitude over short
distances as well as substantial latitudinal and longitudinal variation. A large and complex coast also
introduces considerable climate variability. This makes Norwegian lakes well suited for testing the
effect of climate change on ice phenology, also in relation to altitude.
In the present study, we have analysed long-term (1890-2020) observations of lake freeze-up, ice
break-up and length of ice-free period in 101 Norwegian lakes. The lakes cover a broad range of
climatic zones described by geographical parameters (elevation, latitude and longitude), as well as
lake characteristics (area, water inflow and water level amplitude). The main aim of the analyses was
to detect potential temporal trends in ice phenology while adjusting for both geographical
parameters and lake characteristics.

## 2 Material and methods

### 2.1 Lakes studied

We collated observations from 101 Norwegian lakes, covering a wide range in latitude (58.2 – 69.9
°N), longitude (4.9 – 30.2 °E) and altitude (4 – 1401 m a.s.l.). The lakes are situated in three major
climatic zones (boreal, subalpine, alpine) and with varying distances from the ocean. Thus, they differ
widely in several geographic characteristics (Figure 1, Appendix 1). Most of the lakes are relatively
small (median area 6.9 km$^2$), although the dataset also includes Norway's largest lake, Mjøsa (369.3
km$^2$). Their catchment areas vary between 7.1 and 18101.9 km$^2$ (median 235 km$^2$) and mean annual
inflow to the lakes varies between 5.6 $10^6$ and 9935.7 $10^6$ m$^3$ year$^{-1}$ (median 256 $10^6$ m$^3$ year$^{-1}$). About
50 % of the lakes (N = 53) were developed for hydropower production with an annual water level
variation varying from 1 to 30.3 m. The lake and catchment information were extracted from
www.nve.no.

## 2.2 Ice observations

Observations of the timing of ice formation on the lakes in autumn and ice break-up in spring were
undertaken visually or by fixed-location video cameras. The data were made available by the
Norwegian Water Resources and Energy Directorate (NVE), the hydropower association Glommens
og Laagens Brukseierforening, or by private persons. NVE operates a national hydrological database
that contains information on ice conditions. The first observations are from 1818, but substantial
records started in the 1890s. Video cameras have now replaced visual observations in some lakes.
Satellite data is also being increasingly used to detect ice cover or open water. In our dataset, we
have included lakes with more than 7 years of observations for at least one ice phenology variable in
the analysis. This resulted in 101 lakes of which 76 have a registration period exceeding 30 years
(Figure 2, Appendix 2). The average length of the data series was 53 years (range 11 – 149 years).
The date of ice break-up was set when the lake was estimated to be free of ice based on the available
observations. The length of the ice-free period during summer was then estimated as the difference
between the day of freeze-up in the autumn and the day of ice break-up in spring. All dates are given
as Julian day number during the year (1 January is day 1). For some lakes in certain years ice
formation started in winter after 1 January. For these years the day number was extended past the
normal 365 days. The observations were always made at the same site in each lake. The date of
freeze-up was set when the first formation of ice was observed. Subsequent temporary ice-free
periods, often due to mild weather combined with strong winds, did not change this date. The date
when the whole lake was covered by ice was also noted, when possible. This date is more variable,
and information is frequently missing. It would require extensive travel and several observation
points to ascertain this date with high certainty, unless there are time-lapse cameras or satellite data.
We have a total of 4371 observations on ice break-up, 3035 observations of freeze-up, 4221
observations of when the lakes were completely frozen over, and 2808 observations of the length of
the ice-free period.
Some of the lakes are used as hydropower reservoirs, and thus within-year water level variation may
differ from the normal annual cycle. For such lakes we have included information on the year of
impoundment and the maximum amplitude of water level variation. Although we do not have
information on exact water level variation within a given year, maximum and minimum occurs when
freeze-up and break-up normally take place, respectively.





For one particular large lake there are observations from two different locations (called Tustervatn
and Røssvatn) that were partly overlapping in time. The observations of the time of ice break-up and
ice freeze-up were strongly and positively correlated. The correlation between the two different
estimates of time of freeze-up (r = 0.501, n = 37, p = 0.002) were lower than for the time of break-up
(r = 0.887, n = 38, p < 0.001).  There was no tendency for a particular temporal trend for this
particular lake, so we have used the longest of the two time-series in the analyses.

2.3 Climate data
As a potential large-scale climate driver, especially impacting ice break-up, we used the North
Atlantic Oscillation (NAO) index. We therefore extracted the PCA-based winter (December to March)
NAO index (National Center for Atmospheric Research Staff (Eds.), last modified 10 September 2019:
https://climatedataguide.ucar.edu/climate-data/hurrell-north-atlantic-oscillation-nao-index-pc-
based (accessed 28 October 2020)). Variation in winter NAO is known to impact on winter
temperature and precipitation, depending on location (Hurrell 1995, Stenseth et al. 2003). An
elevated index leads to mild and wet winters in Europe, while a low index leads to cold and dry
winters. The PCA-based winter NAO-index covers the period from 1898 to 2020. The winter index
covers the period December – February, and we used this index to test for large-scale variation in
timing of ice break-up as the winter index influences both winter precipitation and temperature.

2.4 Statistical analyses
2.4.1 Average time of ice break-up and freezing and length of ice-free period
We tested for variation in timing of the different phenological events using general linear models
(glm) and model selection procedures. Based on prior knowledge, we assumed that these timing
traits would vary depending on longitude (Long), latitude (Lat), and elevation above sea level (Alt, m)
and that there might be interactions among these traits. Further, we assumed that distance to the
sea might be important as it impacts on both precipitation and temperature. We estimated the
distance from each lake to the sea as distance from the outlet of the lake to the coastal shelf (a line
drawn between the outermost islands along the coast) on maps (1:1,000,000). An increasing distance
from the coastal shelf line reflects an increasing importance of continental climate. As the coastline
of Norway bends eastwards at increasing latitude, the coastal distance may more correctly reflect
oceanic/continental climate than longitude.



Various lake and catchment characteristics may also have an impact on ice phenology. Thus, in this
analysis we used total lake area (Area, km$^2$), total catchment area (Catch, km$^2$) and annual mean
inflow (Flow, m$^3$) as descriptors.
We started by evaluating the full model including all parameters (Appendix 3 and 4) and performed a
backward selection procedure until we ended with the "best model". Models were compared with
the corrected Akaike Information Criteria (AIC$_c$) (Burnham and Anderson, 1998). Models with AIC$_c$
values 2 units below that of a competing model are assumed to be a better fit to the data. When
presenting the results of the model selection we present the AIC$_c$ values for the three best models as
well as the full model in appendix tables and present the best model by giving parameter estimates
and overall model results.

2.4.2 Temporal variation in timing of ice break-up, freeze up and length of ice-free period
We used several different approaches to test for temporal variation in the different ice phenology
traits.
Firstly, in order to identify the main parameters influencing variation in time of freeze-up, time when
lakes were completely frozen over and length of the ice-free period, we used general linear mixed
models (glmm), using basically the same parameters as in our average modelling approach. Year was,
however, always included as a continuous variable to test for linear temporal trends. In addition, the
parameters Impounded (yes/no) and water level amplitude (Amplitude, m) were always either
excluded or included in parallel in the analyses. To account for temporal autocorrelation of
observations from the same lake we included lake identity as a random factor (random intercept) in
the analyses. We used the same model selection procedure as above, but always kept year as a fixed
factor.
Secondly, to test for temporal variation in timing of ice break-up, we used the same general linear
mixed models, with lake as a random variable (random intercept) and year was always included as a
fixed parameter to test for temporal trends.  To test for which factors influenced the time of ice
break-up, in addition to the year effect, we included a large-scale climate index in the modelling. We
included both a linear and a non-linear effect of NAO as potential drivers of variation in the timing of
ice break-up.  NAO-estimates are only available starting in 1899. Thus, this analysis covers a shorter
time frame than the other traits. We selected the best model based on the AIC criterion (Burnham
and Anderson, 2004).



Thirdly, we wanted to investigate if there has been any non-linearity in the temporal trends.
Numerous papers indicate that large-scale climatic changes have occurred mainly during recent years
(Blenckner et al., 2004; Mishra et al., 2011; Post et al., 2018), especially during the last decades. We
therefore selected several lakes (N = 35) with long and complete data series and analysed for
temporal trends in four different 30-year periods (1900-1930, 1931-1960, 1961-1990, 1991-2020). In
these analyses we applied a simplified approach. We used a general mixed modelling approach, with
ice phenology as response variable, year as predictor, and lake identity as random factor. In these
models we assume that all lakes have the same temporal trends (same slope) within each time
period. Including a random slope did not change the conclusions.
All statistical analyses were performed using JMP 12 (JMP Version 12. SAS Institute Inc., Cary, NC,

208  1989-2019).


## 3 Results

All lakes had distinct periods without ice every year. The observations of average timing of ice break-
up, time of lake freeze-up, time when the lake was completely frozen and length of ice-free period
were strongly correlated (Figure 3, Table 1).

### 3.1 Spatial variation in average ice phenology

We tested for drivers of variation in average time of ice break-up, lake freeze up, time when a lake is
completely frozen over and the length of the ice-free period. A summary of the model selection
results is presented in Appendix 4.
The spatial variation in average time of ice break up was best explained by a complex model including
a three-way interaction between latitude, longitude and altitude (Table 2). The best model did,
however, include a weak negative effect of annual inflow to the lake, but not distance to the sea.
Distance to sea was, however, included in a model within 0.4 $AIC_c$ units of the best model. There
were only small effects of the various lake characteristics, but ice break-up was later with increasing
latitude (2.3 days per °N), longitude (1.5 days per °E) and altitude (3.4 days per 100 m) (Figure 4). The
lakes are situated geographically such that latitude and longitude are strongly positively correlated (r
= 0.825, p< 0.001), indicating that the effects should be interpreted with caution. Furthermore, there
was large within-lake variability in timing of ice break-up (Table 3), with an average coefficient of
variation (CV; defined as standard deviation divided by the mean) of 8.90 %. Within-lake CV was
negatively correlated with latitude, longitude, altitude and distance to the coastline. This indicates
larger phenological variation in lakes in southern and western areas and at lower altitude.
The best models explaining variation in the timing of lake freeze-up, time when the lake is completely
frozen, and the length of the ice-free period usually contained an interaction effect between
longitude and altitude. All models also included a positive effect of lake area (Table 2, Appendix 3).
Overall, lakes freeze up earlier and have a shorter ice-free period with increasing longitude and
altitude. Large lakes also take longer to freeze and were ice-free for longer than smaller lakes. The
within-lake variation in timing of freeze-up (mean CV = 4.45 %) and when the lake was completely
frozen (mean CV = 4.55 %) was less than the variation in the length of the ice-free period (mean CV =
15.04 %). The CV of these three phenological traits were negatively correlated with altitude and
coastal distance (Table 3). The effect of longitude was more variable.

3.2 Temporal variation in timing of lake freeze up, time when the lake is completely
frozen and length of ice-free period
The best models, based on the AIC$_c$ criterion, for timing of lake freeze-up, time when the lake was
completely frozen and the length of the ice-free period contained geographic parameters such as
altitude, latitude and longitude (Appendix 4). Lake area also had a positive effect on all these three
phenological traits. In addition, lake impoundment and the amplitudinal range in water level had an
impact on all traits. There was little temporal variation in these traits on the long timescale analysed
here; only for when the lake was completely frozen over, did we find a significant (p<0.001) positive
temporal trend, indicating that the lakes are completely frozen later in the autumn in recent years
(Table 4).

3.3 Temporal trends in timing of ice break-up
The best model for the timing of ice break-up included the effects of geography, time and climate
(Appendix 5). Ice break-up occurred later during spring with increasing altitude, latitude, and
longitude. These effects are complex, as indicated by the various significant interaction effects. In
addition, there was a significant negative temporal trend in ice break-up, i.e. ice break-up occurred
earlier in the spring (Table 5). There was also a significant climate effect, with a negative linear effect
of the NAO (p<0.001).



3.4 Non-linear temporal trends in ice phenology

Many studies indicate that climate is changing faster during recent decades. To investigate for potential non-linear trends in ice phenology we analysed for temporal trends within four different time periods (1900-1930, 1931-1960, 1961-1990, 1991-2020). We selected 35 lakes with relatively long, and continuous data series exceeding 50 years for both date of break-up and date of completely frozen lake (Appendix 6). We used a period-specific mixed mode, assuming similar temporal trends (slopes) for all lakes (random intercept only). During the three first time periods none of the slope estimates were significant (Figure 5, Table 6), whereas during the last time period (1991-2020) most temporal trends were significant. During this period ice break up happened approximately 2 days earlier per decade, whereas time of ice freeze-up and time when lake is completely frozen were on average 6 and 3 days later per decade. Furthermore, the length of the ice-free period has become 7 days longer per decade, although this effect was marginally non-significant (p = 0.068).

# 4 Discussion

Our analysis of ice phenology of 101 Norwegian lakes covering the period from the 1890s to the present day gave two major results. Firstly, the analysis indicated significant trends in ice phenology in recent years. Ice break-up occurred earlier, ice freeze-up and completely frozen occurred later, and all trends were accelerating. This results in a longer ice-free season. Secondly, the coefficient of variation in the different ice phenology variables were larger in lakes in southern and western areas and at lower altitudes, indicating that lakes in these areas are most influenced by climate change.

## 4.1. Geographical parameters

The investigated lakes cover a range of climatic zones in a latitudinal, longitudinal and elevational perspective. This conglomerate of variables clearly showed complex and significant interactions, especially for ice break-up, indicating the problems in illuminating the individual importance of the geographical parameters. The date of break-up generally increases with latitude, modified by macro-scale circulation, lake characteristic and local circulation (Blenckner et al. 2004, Livingstone et al. 2009). Our results support this latitudinal trend, but we also found that longitude, altitude and lake size had significant effect.

We found that time of ice break-up was delayed by 2.3 days/°N. This is considerably slower than previously documented in Fennoscandia (3.3-5.4 days/°N) (Efremova et al., 2013; Blenckner et al., 2004) and in North America 3.5 days/°N (Williams et al., 2006). There is no obvious reason for this

discrepancy. One possible explanation could be that registration of ice parameters differs both within
and between studies. Moreover, the oceanic effect could modify the relationship as the majority of
lakes in northern Norway are situated close to the ocean in contrast to the southern lakes that are
mostly continental.
Moreover, we found that ice break-up was 3.4 days delayed by a 100 m increase in elevation. This is
also slightly lower than in Karelian lakes where Efremova et al. (2013) found a delay of 5 days/100 m.
Although there is considerable climatic difference between Norway and Karelia as Karelian lakes in
general experience a more continental climate., The Karelian lakes also covers less variation in
altitude.
Although several studies have studied ice phenology in Europe, most of them have not included
longitude in their analyses. On exception is the study of Polish lakes by Wrzesinski et al. (2015). The
lakes are situated in the northern region and covered a wide longitudinal range (14 – 24 °E), although
a somewhat smaller range compared to the Norwegian lakes. Wrzesinski et al. (2015) found that
break-up increased by 1 day/°E, compared to 1.5 days/°E in our study. The location of the Polish lakes
indicate that any effect of the Baltic Ocean is similar. In contrast, the climate becomes more
continental when moving eastwards in Norway, especially south of 61 °N where the mountain chain
that runs north-south creates a distinct difference in climate from west to east. Thus, the longitudinal
effect could as well be due to the climatic conditions as the proximity to the ocean renders the
climate milder in the west. The longitudinal effect should therefore be treated with caution.
However, the global study by Sharma et al. (2019) showed that distance to the coast was important
in determining whether lakes had annual winter ice cover. In our analysis the distance from ocean
did not per se have any significant effect of any of the ice phenology parameters.
Our results demonstrated a complex relationship among geographical parameters describing date of
freeze-up. The best models explaining variation in the timing of lake freeze-up contained an
interaction effect between longitude and altitude, in addition to a positive effect of lake area. This
differs from the results from other studies in the region. The Karelian lakes, covering 54-68 °N,
freeze-up 2.3 days earlier for every degree of latitude (Efremova et al., 2013), while Swedish (58-68
°N) and Finnish (61-69 °N) lakes freeze-up 2.8 and 4.5 days earlier for each degree of latitude,
respectively (Blenckner et al., 2004). The most obvious explanation for this discrepancy is due to
altitudinal variation. The Norwegian lakes cover 1400 m in elevation range, whereas the lakes in
Karelia are all situated lower than 204 m, in Sweden lower than 340 m and in Finland lower than 473
m. An additional complicating factor is the oceanic climate that, if anything, is more pronounced for
Norwegian lakes than lakes in Sweden, Finland and Karelia.



In our model, distance from the coast does not significantly contribute neither to freeze-up nor
break-up date, probably as distance to the coast was included in both in the latitude and longitude
variables. This in in contrast to the analyses of 41 Finnish lakes where a pronounced deflection of
isolines of both freeze-up and break-up date northward near the Baltic Sea coast was documented
(Palecki and Barry, 1986).
The predictable seasonal cycle in solar radiation is characteristic of higher latitudes. Weyhenmeyer et
al. (2011) hypothesised, based on a global dataset, that lakes north of 61 °N had lower inter-annual
variability in seasonal cycle than lakes at latitudes lower than 61 °N. The Norwegian lakes are
distributed along a latitudinal gradient to test this hypothesis in a robust way. Our results lend
support to this, as the within-lake coefficient of variation (CV) of ice break-up, freeze-up and length
of ice-free season were negatively correlated with latitude, longitude, altitude and/or distance to
coastline. This indicates larger phenological variation in lakes in southern and western areas and at
lower altitude.

*4.5 Temporal trends*
Although many studies have documented trends in ice phenology, few studies have investigated
changes across specific periods to elucidate periods with stronger trends. In a study of global
datasets Benson et al. (2012) and Newton and Mullan (2020) showed that trends in ice variables
were steeper over the last 30-year period. Similar increase in trends in the last two decades have
been shown for Karelian lakes (Efremova et al., 2013) and the Great Lakes region (Mishra et al.,

345    2011).

Our analyses revealed significant, accelerating trends for earlier break-up, later freeze-up and
completely frozen lakes after 1991. Moreover, the trend for a longer ice-free period also accelerated
during this period, although the trend was not significant. These trends are in accordance with an
increase in air temperature in the spring and autumn, as well for the global temperature over the last
decades (Benson et al., 2012; Hansen et al., 2006). Our results are in accordance with Newton and
Mullan (2019), showing marked differences in ice phenology in Fennoscandian lakes (Sweden,
Finland) across 30-year periods after 1931. In Newton and Mullan (2020), break-up trends appeared
to be earlier and more pronounced in southern regions during the first period. In the next period,
1961-1999, break-up trends increased in magnitude, and the lakes with negative trends in the
previous period shifted to be positive. In last period, the strength of the trends in earlier break-up
increased and reached 3.9 days/decade. In our study, the trend in the 1991-2020 was 2.0
days/decade. One plausible reason for a slower trend in Norwegian lakes during this period than in
the rest of Fennoscandia is the influence of the ocean. The extension of the Gulf Stream, the North
Atlantic Drift, along the Norwegian coast contributes to a mild climate and reduced climate change
shown by the deflection of the 0 °C winter isotherm going northward (Newton and Mullan 2020).
Moreover, the speed of thermal change in the ocean is less rapid and less variable than in inland
waters (Woolway and Maberly, 2020).
Changes in ice phenology depend on several climatic forcing variables, such as air temperature, solar
radiation, wind and snowfall (Magnusson et al. 1997). A significant increase in global air temperature
during the last century is well documented (e.g. Hansen et al., 2006; Robinson, 2020). Newton and
Mullan (2020) showed that rising temperature appears to be the dominant factor for the shift
towards earlier break-up and later freeze-up in the Northern Hemisphere. Precipitation may also play
a role in the observed trends. Nordli et al. (2007) found a significant correlation ($R^2$=0.58) between
date of break-up in lake Randsfjorden and the mean temperature in February to April. Duguay et al.
(2006) showed that trends towards later freeze-up corresponded with areas of increasing autumn
snow cover, and that spatial trends in break-up were consistent with changes in spring snow cover
duration. Similarly, Jensen et al. (2007) in a study of ice phenology trends across the Laurentian Great
Lakes region found that variability in the strength of trends in earlier break-up were partly explained
by number of snow days or snow depth.  For the lake Litlosvatn, in the mountain area of western
Norway, Borgstrøm (2001) found a clear relationship between spring snow depth and the date on
which the lake was free of ice. The altitudinal gradient causes considerable regional difference in
annual precipitation in Norway (Hanssen-Bauer, 2005). The general trend in increasing temperature
and precipitation observed from 1875 to 2004, has been modelled to increase to 2100, although
there will be regional differences (Hanssen-Bauer et al., 2017). Thus, our results concerning the
recent trends in ice phenology probably indicate a new situation for ice formation in Norwegian
lakes.
Biological consequences
Shifts in ice phenology have major repercussions for the biota of lakes and rivers (Prowse 2001,
Caldwell et al. 2020), as ice cover changes the aquatic environment, not only in terms of light
penetration, but also the physical characteristics of the environment such as temperature. Of special
interest is that the trend in earlier ice break-up and the loss of ice will stimulate biological
production. In late autumn, solar insulation is restricted and thus, a prolonged period without ice has
limited consequences for aquatic production. Caldwell et al. (2020) tested a conceptual model that
expressed how earlier break-up affected aquatic ecosystems. The effect differed between and within
tropic levels. Whereas contrasting effects were found between littoral and pelagic zooplankton
production, the modelled brook trout (*Salvelinus fontinalis*) did not profit from the increased
zooplankton production and experienced reduced fitness. A review of the long-term dynamics of fish
species in Europe (Jeppesen et al. 2011), revealed a shift towards higher dominance of eurythermal
species. Loss of ice cover increased resting metabolism by approximately 30 % in an Atlantic salmon
(*Salmo salar*) population (Finstad et al., 2004), and the recruitment of an alpine brown trout (*Salmo*
*trutta*) population was strongly affected by accumulated snow depth and thereby the timing of ice-
break (Borgstrøm and Museth, 2005). Moreover, the outcome of competition in sympatric
populations of brown trout and Arctic charr (*Salvelinus alpinus*) is strongly dependent on the
duration of ice-cover as high charr abundance is correlated with low trout population growth rate
only in combination with long winters (Helland et al., 2011). In addition, aquatic insects, such as
Ephemeroptera and Plecoptera may change their voltinism and their emergence timing in a warmer
climate (Brittain 1978, 2008; Sand & Brittain 2009). We still have limited knowledge about how
climate change in general may have impacts on Arctic and Alpine fishes and fish populations (Reist et
al., 2006). This is also the case with changes in ice phenology. The biological consequences of
changes in ice phenology will be first and most marked in lakes with high coefficient of variation in
the ice phenology parameters; that is, in lakes situated in the lowlands and in the southern part of
Norway.

## 409 5 Conclusions

Ice phenology is complex and determined by the interaction of a range of parameters. This study
shows that altitude, latitude and longitude all significantly affect ice phenology in Norwegian lakes.
Lake characteristics are of minor importance, although lake size had a significant effect. In addition,
there is a significant temporal effect of changing climate during the most recent time period (1991-
2020). There was a significant trend that lakes were completely frozen over later in the autumn in
recent years, as well as trend for earlier ice break-up in spring. An understanding of the relationship
between ice phenology and geographical and climate parameters is a prerequisite for predicting the
potential consequences of climate change on ice phenology and lake biota.


*Data availability*. All ice phenology data are available at doi:10.5061/dryad.bk3j9kd9x.
*Author contributions*. JHL-L designed this study. JHL-L, LAV and JEB led the writing of this paper. LAV
conducted the formal analysis. Data curation was conducted by JHL-L, ÅSK and TS. JHL-L collated
basic characteristics for individual lakes.



*Competing interests*. The authors declare that they have no conflict of interest.
*Acknowledgements.* We would like to acknowledge Glommens og Laagens Brukseierforening
hydropower company for giving access to ice phenology of 13 lakes. Halvor Lien provided
observation of ice phenology of lake Møsvann which was carried out by Halvor Hamaren until 1987,
and himself afterwards. Julio Pereira, NVE, kindly drew the maps.

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

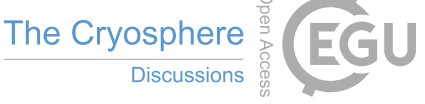


Figure 1. Map showing the locations of the 101 lakes included in the analysis. Information on the
locations and names of the lakes is given in Table S1 in the online Supplement.



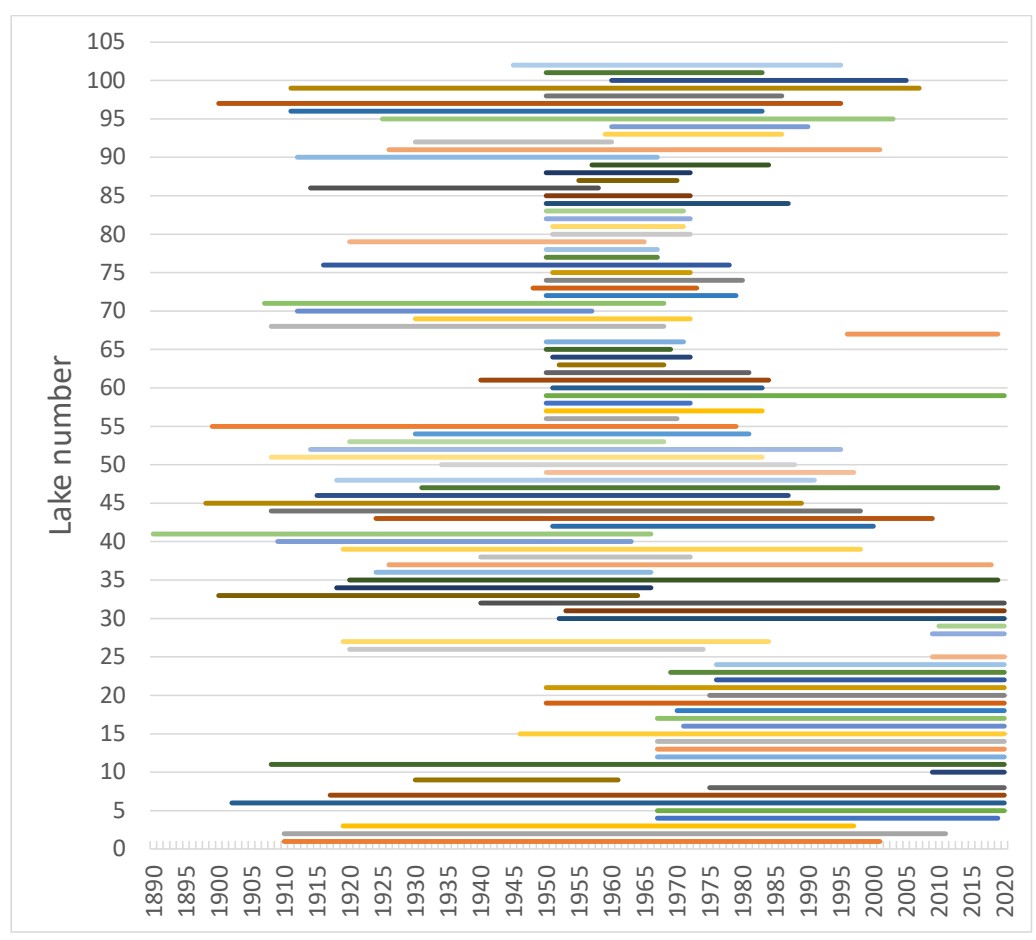


Figure 2. Chart showing the registration periods for ice phenology (ice freeze-up, frozen lake and ice
break-up) for individual lakes. For Lake 41, registration started in 1818 but was not continuous. In
several data series there are years with missing registration of variables. For information on each
lake see Appendix 1.




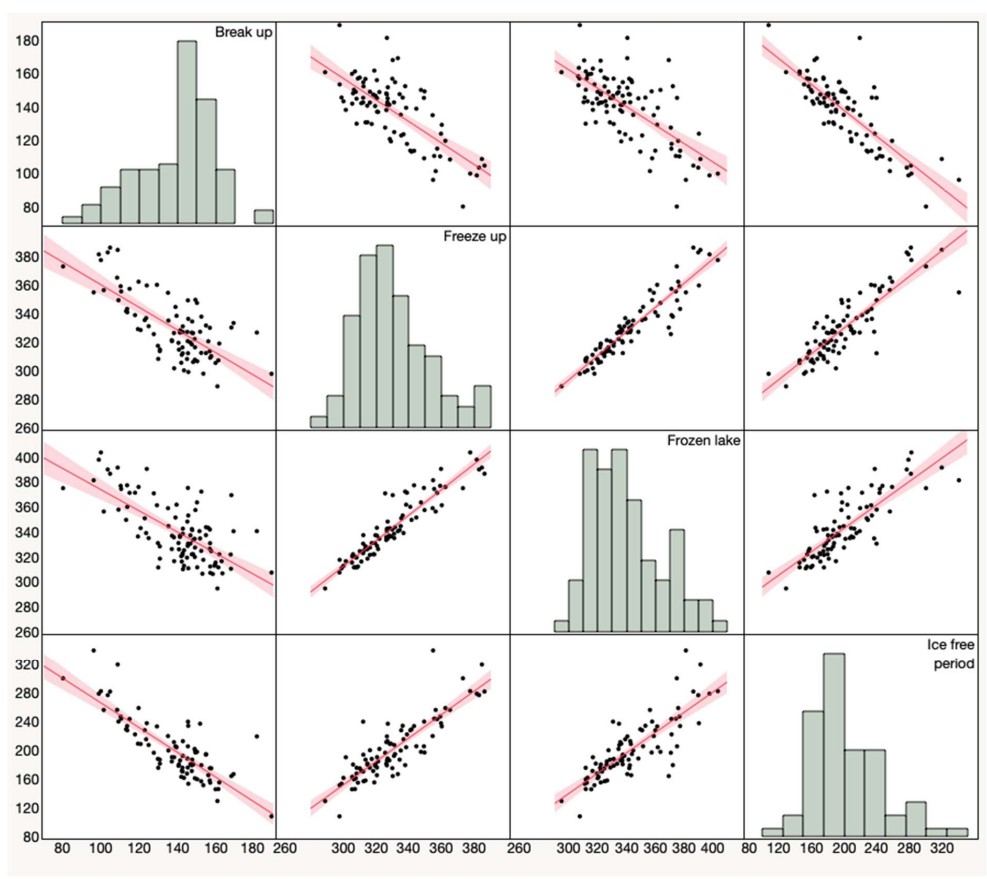


Figure 3. The correlation between the average timing of ice break-up, freeze-up, frozen lake and

length of ice-free period in 101 Norwegian lakes during the period1890-2020.




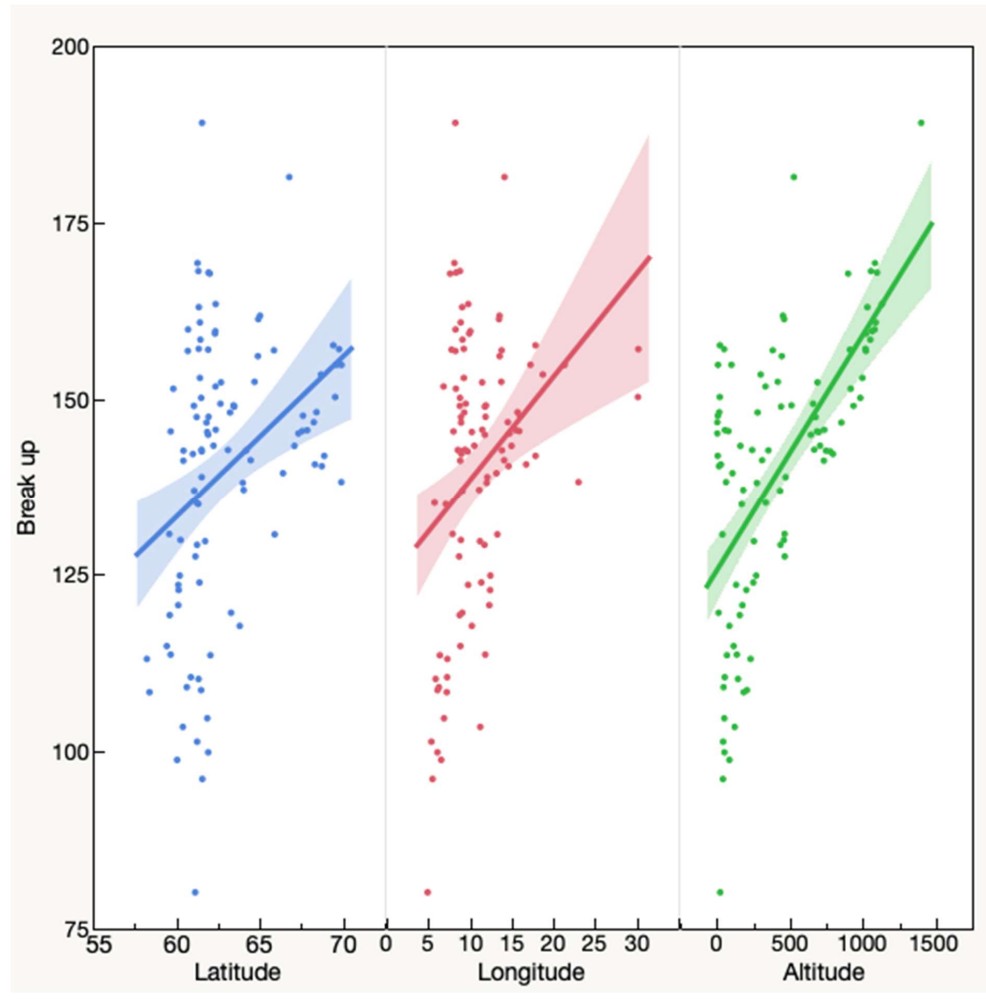


Figure 4. The correlation between the average timing of ice break-up and latitude, longitude and
altitude of 101 Norwegian lakes during the period1890-2020.



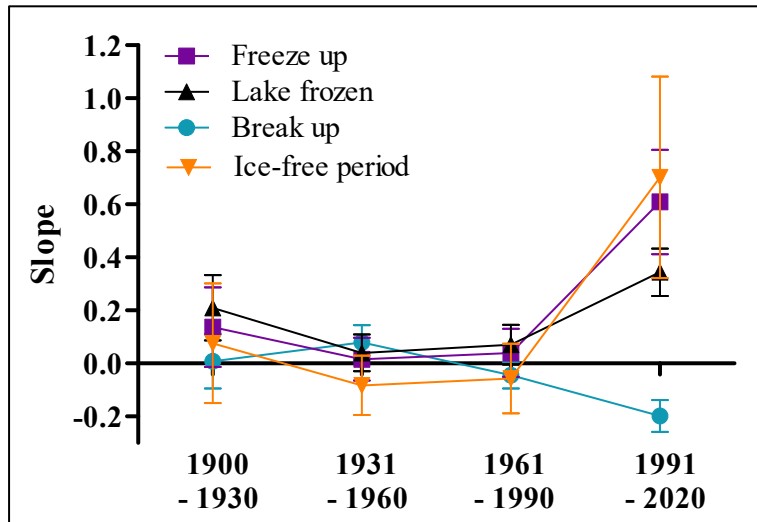


Figure 5. Estimated slopes from general linear mixed models with aspects of ice phenology as
response variables (Parameter estimates and significance level are given in Table 5). Means and
standard deviations are given.



**Table 1.**
Correlation between timing of ice-break-up, lake freeze-up, time when the lake was completely
frozen and length of ice-free period for 101 Norwegian lakes. All correlations coefficients are
significant at P<0.001.

|  | Lake freeze-up | Lake completely frozen | Length of ice-free period |
|---|---|---|---|
| Ice break-up | -0.741 | -0.692 | -0.829 |
| Lake freeze-up |  | 0.934 | 0.868 |
| Lake completely frozen |  |  | 0.829 |







**Table 2. Model summary**. Testing for temporal variation in time of ice break-up, time of lake freeze-
up, time when the lake is completely frozen, and length of ice-free period for 99 lakes in Norway.
Parameter estimates for the best model are given (see Appendix table 1 for results from the model
selection). Significant parameter estimates are given in bold.
**Time of ice break-up**: Summary statistics with parameter estimates ($\beta \pm$ S.E.), *t*-values and
significance level (P). Model *F*-ratio = 91.46 (d.f. = 8, 92), total N = 101, P < 0.0001, $R^2$ = 0.888.

| Parameter | $\beta$ | S.E. | *t*-value | P |
|---|---|---|---|---|
| Intercept | -222.39 | 39.32 | -5.66 | **<0.001** |
| Latitude | 5.58 | 0.69 | 8.08 | **<0.001** |
| Longitude | -0.22 | 0.53 | -0.41 | 0.684 |
| Altitude | 0.36 | 0.004 | 9.41 | **<0.001** |
| Latitude*Longitude | 0.10 | 0.15 | 0.65 | 0.515 |
| Latitude*Altitude | 0.008 | 0.002 | 3.65 | **<0.001** |
| Longitude*Altitude | -0.008 | 0.002 | -4.44 | **<0.001** |
| Latitude*Longitude*Altitude | 0.001 | 0.001 | 2.89 | **0.005** |
| Annual inflow | -0.001 | 0.001 | -1.77 | **0.080** |


**Time of lake freeze up**: Summary statistics with parameter estimates ($\beta \pm$ S.E.), *t*-values and
significance level (P). Model *F*-ratio = 23.14 (d.f. = 6, 80), total N = 87, P < 0.0001, $R^2$ = 0.634.

| Parameter | $\beta$ | S.E. | *t*-value | P |
|---|---|---|---|---|
| Intercept | 394.04 | 64.25 | 6.13 | **<0.001** |
| Latitude | -0.32 | 1.08 | -0.30 | 0.767 |
| Longitude | -3.28 | 0.73 | -4.48 | **<0.001** |
| Altitude | -0.03 | 0.007 | -4.28 | **<0.001** |
| Latitude*Longitude | 0.32 | 0.12 | 2.60 | **0.011** |
| Latitude*Altitude | 0.005 | 0.003 | 1.79 | 0.077 |
| Lake area | 0.14 | 0.03 | 4.05 | **<0.001** |


**Time when lake is completely frozen**: Summary statistics with parameter estimates ($\beta \pm$ S.E.), *t*-
values and significance level (P). Model *F*-ratio = 42.57 (d.f. = 3, 96), total N = 100, P < 0.0001, $R^2$ =

608  0.570.





| Parameter | β | S.E. | t-value | P |
|-----------|------|-------|---------|--------|
| Intercept | 389.92 | 5.66 | 68.84 | **<0.001** |
| Longitude | -3.08 | 0.39 | -7.87 | **<0.001** |
| Altitude | -0.04 | 0.005 | -9.42 | **<0.001** |
| Lake area | 0.15 | 0.04 | 4.12 | **<0.001** |



**Length of ice-free period**: Summary statistics with parameter estimates (β ± S.E.), t-values and
significance level (P). Model F-ratio = 34.06 (d.f. = 6, 80), total N = 87, P < 0.0001, $R^2$ = 0.719.

| Parameter | β | S.E. | t-value | P |
|-----------|------|--------|---------|--------|
| Intercept | 301.63 | 106.90 | 2.82 | **0.006** |
| Latitude | -0.10 | 1.80 | -0.06 | 0.954 |
| Longitude | -6.43 | 1.22 | -5.29 | **<0.001** |
| Altitude | -0.08 | 0.01 | -6.84 | **<0.001** |
| Latitude*Longitude | 0.62 | 0.21 | 3.07 | **0.003** |
| Latitude*Altitude | 0.01 | 0.005 | 1.88 | **0.064** |
| Lake area | 0.15 | 0.06 | 2.73 | **0.008** |




**Tabell 3.** Summary statistics for the coefficient of variation (mean, median and range), and
correlation between CV and various geographic traits for each lake (altitude, latitude, longitude and
distance to the coastline).

| | CV | | | Correlation coefficient | | | |
|---|---|---|---|---|---|---|---|
| | mean | median | range | altitude | latitude | longitude | Coastal distance |
| Ice break-up | 8.94 | 6.87 | 3.94 – 29.93 | -0.477 (<0.001) | -0.238 (0.018) | -0.361 (<0.001) | -0.297 (0.003) |
| Lake freeze-up | 4.45 | 4.16 | 1.94- 10.18 | -0.228 (0.034) | -0.092 (0.397) | -0.229 (0.033) | -0.237 (0.027) |
| Lake completely frozen | 4.60 | 4.31 | 2.82- 9.35 | -0.445 (<0.001) | 0.159 (0.117) | 0.249 (0.808) | -0.367 (<0.001) |
| Length of ice- free period | 15.04 | 11.55 | 5.73- 42.83 | -0.225 (0.036) | 0.542 (<0.001) | 0.324 (0.002) | -0.427 (<0.001) |




**Table 4. Model summary.** Testing for temporal variation in time of lake freeze-up, time when the
lake is completely frozen, and length of ice-free period for 99 lakes in Norway. Lake identity is
modelled as a random factor, and year is always included in the model as a fixed effect. Summary
statistics with parameter estimates (β ± S.E.), *t*-values and significance level (P) for the best model
are given (see Appendix table 2 for results from the model selection). Significant parameter
estimates are given in bold.
**Timing of lake freeze-up:** Total N = 3035, $R^2$ = 0.676, P < 0.0001**.** The random lake effect accounts for
44.0% of total variance**.**

| Parameter | β | S.E. | *t*-value | P |
|---|---|---|---|---|
| Intercept | 491.30 | 62.00 | 7.92 | **<0.001** |
| Year | -0.006 | 0.016 | -0.35 | 0.724 |
| Latitude | -1.82 | 0.92 | -1.97 | 0.052 |
| Longitude | -2.10 | 0.60 | -3.53 | **<0.001** |
| Altitude | -0.04 | 0.005 | -8.10 | **<0.001** |
| Lake area | 0.12 | 0.03 | 3.60 | **<0.001** |
| Impoundment (no) | 0.66 | 0.96 | 0.69 | 0.491 |
| Amplitude | 0.53 | 0.18 | 2.98 | **0.003** |


**Time when lake is completely frozen:** Total N = 4084, $R^2$ = 0.697, P < 0.0001**.** The random lake effect
accounts for 50.6% of total variance**.**

| Parameter | β | S.E. | *t*-value | P |
|---|---|---|---|---|
| Intercept | 301.62 | 65.86 | 4.58 | **<0.001** |
| Year | 0.06 | 0.01 | 4.68 | **<0.001** |
| Latitude | -0.65 | 1.05 | -0.62 | **0.537** |
| Longitude | -2.68 | 0.67 | -4.01 | **<0.001** |
| Altitude | -0.05 | 0.005 | -9.89 | **<0.001** |
| Lake area | 0.15 | 0.04 | 3.93 | **<0.001** |
| Impoundment (no) | -0.53 | 0.84 | -0.63 | 0.526 |
| Amplitude | 0.24 | 0.15 | 1.55 | 0.122 |


**Length of ice-free period:** Total N = 2807, $R^2$ = 0.663, P < 0.0001**.** The random lake effect account for
34.4% of total variance**.**





| Parameter | β | S.E. | *t*-value | P |
|---|---|---|---|---|
| Intercept | 433.89 | 108.63 | 3.99 | **<0.001** |
| Year | 0.02 | 0.03 | 0.52 | 0.606 |
| Latitude | -2.80 | 1.50 | -1.87 | 0.065 |
| Longitude | -6.05 | 1.26 | -4.78 | **<0.001** |
| Altitude | -0.10 | 0.009 | -10.90 | **<0.001** |
| Latitude*Longitude | 0.45 | 0.19 | 2.37 | **0.020** |
| Lake area | 0.16 | 0.06 | 2.87 | **0.005** |
| Impoundment (no) | 4.79 | 1.91 | 2.51 | 0.012 |
| Amplitude | 0.60 | 0.36 | 1.65 | 0.098 |






**Table 5. Model summary.** Temporal and climate effects on in time of ice break-up 98 lakes in
Norway. Lake identity is modelled as a random factor, and year is always included in the model as a
fixed effect. NAO is included as the climate effect. Summary statistics with parameter estimates (β ±
S.E.), *t*-values and significance level (P) for the best model are given (see Appendix table 3 for results
from the model selection). Significant parameter estimates are given in bold.
Total N = 4194, $R^2$ = 0.726, P < 0.0001. The random lake effect account for 22.3 % of total variance.

| Parameter | β | S.E. | *t*-value | P |
|---|---|---|---|---|
| Intercept | -205.98 | 46.00 | -4.42 | **<0.001** |
| NAO | -3.26 | 0.20 | -16.61 | **<0.001** |
| Year | -0.03 | 0.01 | -2.86 | **0.004** |
| Latitude | 6.21 | 0.76 | 8.19 | **<0.001** |
| Longitude | -0.64 | 0.59 | -1.08 | 0.283 |
| Altitude | 0.04 | 0.003 | 13.99 | **<0.001** |
| Latitude * Longitude | -0.30 | 0.07 | -4.35 | **0.004** |
| Latitude * Altitude | 0.008 | 0.002 | 3.59 | **<0.001** |
| Longitude * Altitude | -0.008 | 0.002 | -4.25 | **0.004** |





**Table 6**. Parameters estimates (slope ± se) from general linear mixed models with ice phenology

642         estimates as response variables, year as predictor and lake identity as random effect. The

643         time series are sorted into 30-year periods (1900-1930, 1931-1960, 1961-1990, 1991-2020).

644         Significant estimates are given in bold, with number of observations in parenthesis. The lakes

645         included is given in Appendix


|  | Break up | Freeze up | Lake frozen | Ice-free period |
|---|---|---|---|---|
| 1900-1930 | 0.008±0.102<br>N=392 | 0.137±0.150<br>N=326 | 0.210±0.123<br>N=437 | 0.076 ±0.226<br>N=254 |
| 1931-1960 | 0.080±0.064<br>N=739 | 0.016±0.081<br>N=637 | 0.040±0.069<br>N=734 | -0.083±0.112<br>N=586 |
| 1961-1990 | -0.044±0.050<br>N=772 | 0.040±0.091<br>N=502 | 0.071±0.075<br>N=754 | -0.057±0.1309<br>N=475 |
| 1991-2020 | **-0.198±0.060**<br>**N=411** | **0.609±0.197**<br>**N=116** | **0.344±0.089**<br>**N=391** | 0.702±0.380<br>N=107 |







**Appendix 1.**
Lake characteristics of the 101 Norwegian lakes used in the analyses.

| Lake no | Lake | North | East | Coastal distance (km) | Altitude (m asl.) | Area (km2) | Mean annual inflow (10exp6 m3) | Catchment (km2) | Impounded |
|---|---|---|---|---|---|---|---|---|---|
| 1 | Mjøsa (Hamar) | 60,397 | 11,234 | 350 | 123 | 369,32 | 9953,72 | 16555,36 | 1920 (3.61 m) |
| 2 | Storsjø | 61,392 | 11,363 | 357 | 251 | 48,1 | 1027,59 | 2293,6 | 1968 (3.64 m) |
| 3 | Lomnessjøen | 61,732 | 11,202 | 329 | 255 | 3,67 | 511,93 | 1164,41 | no |
| 4 | Osensjøen | 61,246 | 11,739 | 385 | 437 | 43,37 | 665,79 | 1174,36 | 1941 (6.6 m) |
| 5 | Olstappen | 61,514 | 9,402 | 231 | 668 | 3,2 | 1188,82 | 1305,11 | 1954 (13 m) |
| 6 | Aursunden | 62,68 | 11,462 | 196 | 690 | 46,11 | 629,99 | 848,44 | 1923 (5.9 m) |
| 7 | Atnsjøen | 61,852 | 10,226 | 217 | 701 | 5,01 | 323,1 | 463,2 | no |
| 8 | Savalen | 62,232 | 10,519 | 189 | 708 | 15,29 | 29,93 | 102,48 | 1973 (4.7 m) |
| 9 | Narsjø | 62,364 | 11,477 | 238 | 737 | 1,95 | 70,67 | 118,86 | no |
| 10 | Gålåvatn | 61,53 | 9,717 | 270 | 778 | 3,04 | 9,72 | 23,1 | no |
| 11 | Tesse | 61,814 | 8,941 | 182 | 854 | 12,84 | 102,24 | 225,37 | 1942 (12 m) |
| 12 | Aursjø | 61,934 | 8,327 | 140 | 1098 | 6,7 | 41,61 | 106,31 | 1967 (14.5 m) |
| 13 | Breidalsvatn | 62,008 | 7,63 | 123 | 900 | 6,9 | 177,02 | 127,22 | 1944 (13 m) |
| 14 | Raudalsvatn | 61,911 | 7,796 | 109 | 913 | 7,48 | 209,08 | 146,93 | 1952 (30.3 m) |
| 15 | Gjende | 61,495 | 8,81 | 196 | 984 | 15,61 | 497,31 | 376,2 | no |
| 16 | Veslevatn | 61,416 | 9,273 | 224 | 998 | 4,22 | 33,98 | 44,11 | 1960 (2 m) |
| 17 | Kaldfjorden | 61,35 | 9,263 | 245 | 1019 | 19,18 | 655,29 | 559,88 | 1956 (4.9 m) |
| 18 | Fundin | 62,324 | 9,915 | 161 | 1022 | 10,4 | 155,13 | 252,86 | 1968 (11 m) |
| 19 | Vinstern | 61,352 | 9,069 | 238 | 1032 | 28,19 | 573,95 | 466,3 | 1951 (4 m) |
| 20 | Nedre Heimdalsvatn | 61,446 | 9,108 | 238 | 1052 | 7,25 | 134,72 | 129,2 | 1959 (2.2 m) |
| 21 | Bygdin | 61,328 | 8,799 | 235 | 1057 | 40,03 | 398,02 | 305,59 | 1934 (9.15) |
| 22 | Marsjø | 62,343 | 10,049 | 165 | 1064 | 2,68 | 13,95 | 23,39 | 1910 (4 m) |
| 23 | Øvre Heimdalsvatn | 61,418 | 8,893 | 203 | 1089 | 0,78 | 26,89 | 24,94 | no |
| 24 | Elgsjø | 62,361 | 9,798 | 154 | 1132 | 2,38 | 22,16 | 33,75 | 1914 (5.35 m) |
| 25 | Leirvatnet | 61,547 | 8,25 | 168 | 1401 | 1,04 | 170,31 | 154,72 | no |
| 26 | Volbufjorden | 61,08 | 9,11 | 238 | 434 | 3,94 | 446,88 | 675,85 | 1916 (3 m) |
| 27 | Øyangen | 61,221 | 8,924 | 231 | 677 | 6,64 | 238,64 | 246,19 | 1918 (8.3 m) |
| 28 | Vasetvatnet | 60,996 | 8,985 | 231 | 796 | 1,03 | 47,81 | 82,9 | no |
| 29 | Midtre Syndin | 61,058 | 8,782 | 224 | 937 | 2,73 | 15,68 | 21,47 | no |
| 30 | Rødungen | 60,696 | 8,256 | 193 | 1022 | 7,4 | 51,01 | 61,79 | 1943 (23 m) |
| 31 | Bergsjø | 60,709 | 8,275 | 193 | 1082 | 1,68 | 5,58 | 28,09 | 1943 (11 m) |
| 32 | Vangsmjøsa | 61,149 | 8,701 | 231 | 466 | 17,4 | 22,97 | 487,6 | 1963 (3 m) |
| 33 | Krøderen | 60,123 | 9,783 | 270 | 133 | 43,91 | 3701,57 | 5091,06 | 1960 (2.6 m) |
| 34 | Fønnebøfjorden | 60,256 | 8,914 | 217 | 460 | 0,75 | 455,12 | 687,29 | no |
| 35 | Tunhovdfjorden | 60,426 | 8,833 | 221 | 734 | 25,55 | 1141,64 | 1857,98 | 1920 (18.15 m) |
| 36 | Pålsbufjorden | 60,433 | 8,733 | 215 | 749 | 19,64 | 1063,35 | 1645,84 | 1946 (24.5 m) |





| | | | | | | | | |
|---|---|---|---|---|---|---|---|---|
| 37 | Møsvatn | 59,824 | 8,317 | 182 | 918 | 78,51 | 1573,04 | 1509,77 | 1903 (18.5) |
| 38 | Seljordvatn | 59,434 | 8,854 | 214 | 116 | 16,49 | 428,07 | 724,97 | 1943 (1 m) |
| 39 | Hjartsjå | 59,608 | 8,763 | 210 | 158 | 1,07 | 185,76 | 214,35 | 1957 (1.8 m) |
| 40 | Vinjevatn | 59,582 | 7,926 | 158 | 465 | 3,32 | 1249,03 | 905,89 | 1960 (3.5 m) |
| 41 | Totak | 59,664 | 8,026 | 168 | 687 | 36,59 | 1005,39 | 863,22 | 1958 (7 m) |
| 42 | Eptevatn | 58,236 | 7,291 | 34 | 232 | 1,16 | 51,82 | 33,49 | 1921 (10 m) |
| 43 | Lygne | 58,397 | 7,221 | 53 | 185 | 7,71 | 525,4 | 272,2 | no |
| 44 | Sandvinvatn | 60,053 | 6,555 | 91 | 87 | 4,37 | 1288,75 | 470,22 | no |
| 45 | Vangsvatn | 60,63 | 6,277 | 88 | 47 | 7,65 | 2225,36 | 1091,51 | no |
| 46 | Vassbygdvatn | 60,876 | 7,264 | 147 | 55 | 1,85 | 1136,22 | 760,47 | 1982 (1.4 m) |
| 47 | Tyin | 61,275 | 8,139 | 189 | 1084 | 33,21 | 241,97 | 183,45 | 1942 (10.3 m) |
| 48 | Veitastrondvatn | 61,322 | 7,11 | 133 | 171 | 17,46 | 895,59 | 386,46 | 1982 (2.5 m) |
| 49 | Rørvikvatn | 61,208 | 5,761 | 62 | 336 | 7,14 | 59,9 | 20,69 | 1920 (1 m) |
| 50 | Hersvikvatn | 61,135 | 4,929 | 17 | 24 | 1,37 | 13,53 | 7,06 | no |
| 51 | Nautsundvatn | 61,252 | 5,379 | 39 | 44 | 0,676 | 595 | 218,87 | no |
| 52 | Hestadfjorden | 61,335 | 5,887 | 67 | 146 | 3,24 | 1351,35 | 507,94 | no |
| 53 | Jølstervatn | 61,492 | 6,113 | 77 | 207 | 39,24 | 928,16 | 384,54 | 1952 (1.25m) |
| 54 | Blåmannsvatn | 61,562 | 5,517 | 44 | 43 | 0,24 | 624,99 | 225,49 | no |
| 55 | Lovatn | 61,86 | 6,89 | 98 | 52 | 10,7 | 479,49 | 234,88 | no |
| 56 | Hornindalsvatn | 61,916 | 6,109 | 58 | 53 | 19,09 | 727,73 | 381,04 | no |
| 57 | Kaldvatn | 62,045 | 6,395 | 59 | 70 | 0,78 | 95,7 | 62,02 | 1955 (3 m) |
| 58 | Nysetervatn | 62,352 | 6,835 | 55 | 334 | 2,36 | 59,93 | 29,65 | 1955 (13 m) |
| 59 | Gjevilvatn | 62,648 | 9,49 | 112 | 660 | 21,18 | 167,83 | 169,63 | 1973 (15 m) |
| 60 | Engelivatn | 63,1 | 8,545 | 56 | 243 | 1,81 | 41,51 | 20,6 | 1942 (7.5 m) |
| 61 | Søvatn | 63,226 | 9,308 | 70 | 280 | 5,17 | 156,64 | 101,44 | 1940 (19.8 m) |
| 62 | Rovatn | 63,287 | 9,069 | 560 | 13 | 7,74 | 352,35 | 237,87 | no |
| 63 | Fjergen | 63,434 | 11,91 | 126 | 512 | 13,45 | 303,99 | 227,42 | 1993 (16 m) |
| 64 | Funnsjøen | 63,48 | 11,787 | 119 | 441 | 7,99 | 82,07 | 60,91 | 1938 (11.5 m) |
| 65 | Lustadvatn | 63,991 | 12,013 | 91 | 275 | 7,11 | 82,46 | 68,81 | no |
| 66 | Follavatn | 64,04 | 11,113 | 53 | 182 | 1,44 | 420,12 | 252,29 | 1923 (9.5 m) |
| 67 | Krinsvatn | 63,804 | 10,227 | 35 | 87 | 0,41 | 413,8 | 205,67 | no |
| 68 | Namsvatn | 65,019 | 13,539 | 98 | 454 | 39,44 | 1009,35 | 700,8 | 1951 (14 m) |
| 69 | Fustvatn | 65,899 | 13,286 | 70 | 39 | 16,65 | 970,52 | 475,8 | No |
| 70 | Røssvatn | 65,858 | 13,794 | 91 | 384 | 47,78 | 2513,59 | 1501,21 | No |
| 71 | Tustervatn | 65,858 | 13,794 | 91 | 384 | 47,78 | 2513,59 | 1501,21 | 1957 (13 m) |
| 72 | Vassvatn | 66,397 | 13,176 | 35 | 108 | 0,81 | 66,17 | 16,39 | No |
| 73 | Storglåmvatn | 66,773 | 14,143 | 49 | 529 | 6,18 | 72,53 | 84,79 | 1964 (12.5 m) |
| 74 | Skarsvatn | 67,084 | 14,982 | 56 | 162 | 0,29 | 164,97 | 145,08 | No |
| 75 | Vatnevatn | 67,32 | 14,75 | 35 | 4 | 6,64 | 196,07 | 141,18 | No |
| 76 | Kobbvatn | 67,597 | 15,97 | 70 | 8 | 4,9 | 782,19 | 387,22 | No |
| 77 | Sørfjordvatn | 67,549 | 15,901 | 70 | 80 | 0,31 | 212,45 | 116 | No |
| 78 | Storvatn | 67,848 | 15,503 | 35 | 56 | 6,6 | 155,58 | 71,28 | No |



| 79 | Forsavatn | 68,31 | 16,739 | 112 | 29 | 1,2 | 250,48 | 232,54 | No |
| 80 | Sneisvatn | 68,405 | 15,709 | 74 | 17 | 0,37 | 86,75 | 29,45 | No |
| 81 | Svolværvatn | 68,246 | 14,541 | 21 | 4 | 0,93 | 21,45 | 18,5 | No |
| 82 | Gåslandsvatn | 68,723 | 14,628 | 140 | 16 | 1,54 | 11,9 | 7,35 | No |
| 83 | Skodbergvatn | 68,62 | 17,252 | 91 | 101 | 8,56 | 128,92 | 107,41 | 1953 (6.5 m) |
| 84 | Nervatn | 68,869 | 17,867 | 77 | 7 | 1,2 | 681,76 | 535,57 | No |
| 85 | Lysevatn | 69,413 | 17,86 | 28 | 22 | 41,94 | 281,02 | 129,46 | No |
| 86 | Insetvatn | 68,677 | 18,735 | 126 | 301 | 3,72 | 1267,32 | 1389,68 | No |
| 87 | Oksfjordvatn | 69,903 | 21,347 | 56 | 9 | 58,12 | 256,65 | 265,83 | No |
| 88 | Lille Mattisvatn | 69,894 | 23,016 | 102 | 64 | 11,12 | 267,81 | 318,95 | No |
| 89 | Lille Ropelvvann | 69,761 | 30,188 | 18 | 51 | 1,19 | 20,41 | 48,87 | No |
| 90 | Bjørnvatn | 69,527 | 30,139 | 41 | 21 | 3,54 | 5207,35 | 18101,09 | No |
| 91 | Murusjøen | 64,46 | 14,103 | 168 | 311 | 7,19 | 266,73 | 346,39 | No |
| 92 | Limingen | 64,693 | 13,76 | 140 | 418 | 95,7 | 746,52 | 673 | 1955 (9 m) |
| 93 | Vekteren | 64,894 | 13,563 | 119 | 446 | 8,8 | 381,72 | 310,05 | 1963 (5.5 m) |
| 94 | Saksvatn | 64,919 | 13,482 | 112 | 462 | 1,69 | 76,14 | 63,86 | No |
| 95 | Lenglingen | 64,196 | 13,83 | 168 | 354 | 30,26 | 467,61 | 452,54 | No |
| 96 | Engeren | 61,527 | 12,082 | 364 | 472 | 11,49 | 231,52 | 395,05 | No |
| 97 | Femunden | 61,935 | 11,868 | 336 | 664 | 203,4 | 807,97 | 1793,94 | No |
| 98 | Isteren | 61,91 | 11,779 | 340 | 645 | 80,64 | 1129,71 | 2445,91 | No |
| 99 | Møkeren | 60,12 | 12,318 | 406 | 176 | 12,77 | 75,24 | 367,63 | 1928 (1.2 m) |
| 100 | Søndre Øyersjøen | 60,209 | 12,448 | 417 | 270 | 2,06 | 34,26 | 66,26 | 1934 (4 m) |
| 101 | Varalden | 60,144 | 12,416 | 413 | 203 | 6,5 | 103,95 | 214,11 | 1929 (4.5 m) |
| 102 | Rømsjøen | 59,665 | 11,836 | 385 | 138 | 13,66 | 65,28 | 91,89 | No |





**Appendix 2.**

Summary of ice phenology recordings from 101 Norwegian lakes. Minimum and maximum recordings
are given in brackets.

| Lake no | Lake | Period | n (Break up) | Median (Break up) | n (Freeze up) | Median (Freeze up) | n (Frozen lake) | Median (Frozen lake) | n (Ice free period) | Median (Ice free period) |
|---|---|---|---|---|---|---|---|---|---|---|
| 1 | Mjøsa (Hamar) | 1910-2001 | 76 | 111 (23-139) | 74 | 383 (318-440) | 63 | 392 (350-435) | 63 | 272 (208-401) |
| 2 | Storsjø | 1910-2011 | 66 | 124 (97-140) | 48 | 361 (333-392) | 76 | 390 (349-443) | 28 | 239 (200-276) |
| 3 | Lomnessjøen | 1919-1997 | 66 | 131 (96-147) | 69 | 320 (281-352) | 54 | 327 (302-379) | 58 | 186 (152-248) |
| 4 | Osensjøen | 1967-2019 | 49 | 130 (106-142) | 24 | 360 (336-394) | 50 | 362 (338-406) | 22 | 232 (210-277) |
| 5 | Olstappen | 1967-2020 | 53 | 142 (129-158) | | | 52 | 309 (285-329) | | |
| 6 | Aursunden | 1902-2020 | 115 | 152 (129-175) | 58 | 314 (295-332) | 116 | 324 (295-355) | 57 | 158 (127-186) |
| 7 | Atnsjøen | 1917-2020 | 87 | 145 (122-165) | 95 | 320 (302-347) | 98 | 328 (312-363) | 84 | 176 (144-213) |
| 8 | Savalen | 1975-2020 | 45 | 144 (128-160) | | | 45 | 323 (306-360) | | |
| 9 | Narsjø | 1930-1961 | 31 | 145 (136-164) | 29 | 300 (283-313) | 31 | 311 (293-335) | 29 | 154 (125-175) |
| 10 | Gålåvatn | 2009-2020 | 11 | 145 (124-150) | 11 | 315 (305-326) | 11 | 322 (305-339) | 10 | 175 (162-196) |
| 11 | Tesse | 1908-2020 | 74 | 148 (121-167) | | | 76 | 330 (311-363) | | |
| 12 | Aursjø | 1967-2020 | 53 | 169 (148-181) | | | 53 | 310 (293-332) | | |
| 13 | Breidalsvatn | 1967-2020 | 53 | 168 (147-191) | | | 53 | 323 (303-347) | | |
| 14 | Raudalsvatn | 1967-2020 | 53 | 157 (136-176) | | | 53 | 329 (313-365) | | |
| 15 | Gjende | 1946-2020 | 15 | 149 (137-161) | 14 | 348 (326-377) | 19 | 358 (335-412) | 12 | 194 (175-225) |
| 16 | Veslevatn | 1971-2018 | 47 | 153 (84-182) | | | 47 | 305 (285-332) | | |
| 17 | Kaldfjorden | 1967-2020 | 53 | 159 (136-170) | | | 53 | 309 (285-332) | | |
| 18 | Fundin | 1970-2020 | 50 | 159 (138-174) | | | 48 | 313 (297-328) | | |
| 19 | Vinstern | 1950-2020 | 64 | 163 (147-181) | | | 69 | 317 (288-339) | | |
| 20 | Nedre Heimdalsvatn | 1975-2020 | 45 | 159 (134-171) | | | 45 | 308 (283-326) | | |
| 21 | Bygdin | 1950-2020 | 64 | 170 (153-185) | 15 | 326 (301-382) | 65 | 370 (315-416) | 14 | 157 (130-221) |
| 22 | Marsjø | 1976-2020 | 45 | 160 (135-180) | | | 44 | 314 (297-328) | | |
| 23 | Øvre Heimdalsvatn | 1969-2020 | 49 | 161 (137-188) | 12 | 289 (277-302) | 39 | 294 (279-309) | 12 | 128 (111-151) |
| 24 | Elgsjø | 1976-2020 | 45 | 164 (144-180) | | | 44 | 306 (291-328) | | |
| 25 | Leirvatnet | 2009-2020 | 11 | 182 (157-234) | 11 | 299 (283-312) | 11 | 308 (286-331) | 10 | 120 (55-142) |
| 26 | Volbufjorden | 1920-1974 | 55 | 137 (119-150) | 54 | 320 (305-344) | 55 | 324 (312-353) | 54 | 184 (164-214) |
| 27 | Øyangen | 1919-1984 | 65 | 149 (130-168) | 62 | 318 (299-343) | 62 | 321 (304-344) | 61 | 170 (137-200) |
| 28 | Vasetvatnet | 2009-2020 | 11 | 143 (122-152) | 11 | 307 (294-361) | 11 | 315 (295-363) | 10 | 163 (151-218) |
| 29 | Midtre Syndin | 2010-2020 | 10 | 150 (128-158) | 9 | 309 (280-332) | 10 | 320 (302-334) | 8 | 156 (129-187) |
| 30 | Rødungen | 1952-2020 | 41 | 157 (112-175) | 37 | 312 (301-335) | 47 | 324 (311-366) | 31 | 154 (136-223) |
| 31 | Bergsjø | 1953-2020 | 58 | 160 (146-175) | 47 | 304 (288-343) | 56 | 314 (294-350) | 47 | 144 (127-170) |
| 32 | Vangsmjøsa | 1940-2020 | 34 | 134 (78-149) | 33 | 323 (303-366) | 32 | 375 (315-409) | 32 | 196 (161-276) |
| 33 | Krøderen | 1900-1964 | 64 | 124 (100-161) | 7 | 335 (315-366) | 60 | 338 (306-372) | 7 | 214 (189-255) |
| 34 | Fønnebøfjorden | 1918-1966 | 44 | 131 (104-145) | 15 | 310 (290-321) | 47 | 309 (289-366) | 15 | 174 (152-201) |
| 35 | Tunhovdfjorden | 1920-2020 | 73 | 142 (119-161) | 45 | 329 (275-353) | 77 | 335 (305-362) | 41 | 186 (142-219) |





| 36 | Pålsbufjorden | 1924-1966 | 37 | 145 (121-153) | 32 | 310 (294-355) | 39 | 321 (305-424) | 31 | 166 (143-279) |
|----|---------------|-----------|----|----|----|----|----|----|----|----|
| 37 | Møsvatn | 1926-2018 | 86 | 152 (134-176) | | | 30 | 341 (319-360) | | |
| 38 | Seljordvatn | 1940-1972 | 30 | 115 (89-132) | 26 | 359 (322-386) | 23 | 367 (349-413) | 24 | 244 (279-211) |
| 39 | Hjartsjå | 1919-1998 | 74 | 121 (91-139) | 43 | 328 (311-354) | 70 | 334 (313-388) | 42 | 207 (184-261) |
| 40 | Vinjevatn | 1909-1963 | 46 | 133 (103-146) | 16 | 313 (296-344) | 46 | 317 (297-375) | 16 | 182 (150-220) |
| 41 | Totak | 1818-1966 | 79 | 146 (124-169) | 25 | 348 (332-371) | 20 | 373 (349-408) | 22 | 207 (186-230) |
| 42 | Eptevatn | 1951-2000 | 45 | 114 (22-136) | 36 | 340 (315-382) | 49 | 346 (327-386) | 32 | 224 (182-318) |
| 43 | Lygne | 1924-2009 | 72 | 112 (22-137) | 71 | 362 (441-313) | | | 60 | 253 (212-363) |
| 44 | Sandvinvatn | 1908-1998 | 59 | 106 (33-131) | 61 | 383 (224-437) | 64 | 398 (359-453) | 46 | 276 (225-342) |
| 45 | Vangsvatn | 1898-1989 | 69 | 113 (38-138) | 46 | 347 (316-402) | 78 | 354 (327-420) | 61 | 236 (197-333) |
| 46 | Vassbygdvatn | 1915-1987 | 69 | 116 (56-139) | 56 | 356 (277-401) | 65 | 371 (330-435) | 54 | 242 (158-305) |
| 47 | Tyin | 1931-2019 | 26 | 170 (148-198) | 29 | 335 (314-372) | 30 | 338 (318-373) | 24 | 166 (128-208) |
| 48 | Veitastrondvatn | 1918-1991 | 65 | 137 (76-152) | 52 | 353 (311-416) | 61 | 356 (326-428) | 50 | 217 (171-284) |
| 49 | Rørvikvatn | 1950-1997 | 47 | 137 (91-166) | 47 | 335 (374-310) | 48 | 342 (322-397) | 46 | 199 (159-236) |
| 50 | Hersvikvatn | 1934-1988 | 45 | 83 (18-115) | 47 | 370 (335-413) | 42 | 372 (337-412) | 47 | 292 (245-395) |
| 51 | Nautsundvatn | 1908-1983 | 55 | 106 (33-130) | 75 | 353 (314-426) | 75 | 353 (314-426) | 54 | 248 (215-348) |
| 52 | Hestadfjorden | 1914-1995 | 70 | 117 (17-140) | 75 | 358 (320-423) | 77 | 371 (323-446) | 65 | 242 (192-382) |
| 53 | Jølstervatn | 1920-1968 | 22 | 112 (67-137) | 24 | 384 (340-434) | 12 | 392 (352-430) | 24 | 310 (235-406) |
| 54 | Blåmannsvatn | 1930-1981 | 15 | 95 (39-122) | 40 | 348 (323-407) | 39 | 380 (332-436) | 40 | 347 (221-407) |
| 55 | Lovatn | 1899-1979 | 72 | 108 (18-132) | 44 | 388 (347-436) | 51 | 388 (355-440) | 42 | 281 (227-395) |
| 56 | Hornindalsvatn | 1950-1970 | 20 | 105 (58-128) | 19 | 371 (359-414) | 8 | 406 (378-422) | 19 | 275 (232-363) |
| 57 | Kaldvatn | 1950-1983 | 33 | 113 (82-135) | 32 | 342 (314-385) | 28 | 373 (340-423) | 32 | 228 (179-341) |
| 58 | Nysetervatn | 1950-1972 | 16 | 145 (120-180) | 13 | 324 (309-376) | 17 | 331 (312-381) | 13 | 190 (163-329) |
| 59 | Gjevilvatn | 1950-2020 | 13 | 151 (133-163) | 15 | 347 (321-377) | 18 | 356 (323-387) | 8 | 194 (171-226) |
| 60 | Engelivatn | 1951-1983 | 24 | 144 (118-158) | 25 | 343 (298-344) | 27 | 343 (321-368) | 25 | 186 (147-344) |
| 61 | Søvatn | 1940-1984 | 44 | 146 (118-250) | 42 | 325 (308-347) | 42 | 332 (313-362) | 41 | 180 (62-229) |
| 62 | Rovatn | 1950-1981 | 28 | 126 875)-135) | 31 | 361 (325-413) | 31 | 374 (341-416) | 27 | 235 (200-302) |
| 63 | Fjergen | 1952-1968 | 28 | 152 (122-160) | 27 | 318 (294-335) | 34 | 325 (309-366) | 21 | 166 (141-191) |
| 64 | Funnsjøen | 1951-1972 | 18 | 151 (131-169) | 21 | 322 (297-341) | 18 | 335 (310-362) | 17 | 177 (141-204) |
| 65 | Lustadvatn | 1950-1969 | 13 | 140 (127-147) | 12 | 327 (306-341) | 17 | 338 (314-353) | 10 | 194 (164-210) |
| 66 | Follavatn | 1950-1971 | 20 | 138 (118-155) | 19 | 321 (303-343) | 20 | 333 (312-367) | 18 | 178 (163-222) |
| 67 | Krinsvatn | 1996-2019 | 16 | 119 (92-134) | 15 | 340 (246-384) | 16 | 367 (327-4379 | 11 | 224 (181-262) |
| 68 | Namsvatn | 1908-1968 | 57 | 163 (137-184) | 19 | 319 (301-341) | 58 | 323 (291-351) | 17 | 164 (126-183) |
| 69 | Fustvatn | 1930-1972 | 34 | 135 (84-162) | 37 | 315 (280-347) | 39 | 329 (288-372) | 30 | 182 (151-249) |
| 70 | Røssvatn | 1912-1957 | 46 | 160 (141-182) | 44 | 354 (310-406) | 45 | 370 (337-417) | 44 | 198 (144-248) |
| 71 | Tustervatn | 1907-1968 | 54 | 156 (137-178) | 44 | 328 (304-366) | 50 | 343 (308-391) | 41 | 174 (127-216) |
| 72 | Vassvatn | 1950-1979 | 29 | 137 (113-175) | 29 | 340 (314-363) | 29 | 361 (330-412) | 28 | 199 (158-232) |
| 73 | Storglåmvatn | 1948-1973 | 17 | 178 (162-210) | 20 | 329 (288-361) | 20 | 342 (294-391) | 12 | 151 (89-187) |
| 74 | Skarsvatn | 1950-1980 | 30 | 144 (124-165) | 29 | 300 (278-320) | 30 | 310 (286-355) | 28 | 154 (129-183) |
| 75 | Vatnevatn | 1951-1972 | 21 | 141 (127-247) | 21 | 325 (300-351) | 20 | 345 (325-373) | 20 | 186 (85-214) |
| 76 | Kobbvatn | 1916-1978 | 61 | 149 (128-167) | 58 | 330 (304-386) | 60 | 339 (310-392) | 56 | 185 (140-245) |
| 77 | Sørfjordvatn | 1950-1967 | 7 | 145 (133-159) | 10 | 310 (294-331) | 15 | 322 (306-384) | 5 | 171 (152-191) |





| 78 | Storvatn | 1950-1967 | 9 | 144 (134-158) | 10 | 335 (305-373) | 14 | 353 (334-376) | 7 | 190 (149-239) |
| 79 | Forsavatn | 1920-1965 | 44 | 141 (117-159) | 46 | 316 (288-350) | 46 | 325 (294-403) | 44 | 176 (134-206) |
| 80 | Sneisvatn | 1951-1972 | 19 | 147 (129-180) | 20 | 303 (274-340) | 21 | 311 (284-395) | 17 | 156 (121-194) |
| 81 | Svolværvatn | 1951-1971 | 19 | 149 (116-168) | 20 | 318 (297-318) | 20 | 334 (310-393) | 19 | 173 (135-206) |
| 82 | Gåslandsvatn | 1950-1972 | 22 | 141 (111-172) | 21 | 316 (283-359) | 22 | 327 (309-367) | 21 | 179 (128-229) |
| 83 | Skodbergvatn | 1950-1971 | 11 | 151 (146-167) | 9 | 331 (317-361) | 9 | 336 (329-362) | 8 | 177 (152-212) |
| 84 | Nervatn | 1950-1987 | 37 | 144 (104-160) | 36 | 305 (289-324) | 36 | 319 (301-362) | 34 | 164 (142-192) |
| 85 | Lysevatn | 1950-1972 | 19 | 158 (127-177) | 20 | 307 (296-332) | 20 | 323 (305-359) | 17 | 152 (125-170) |
| 86 | Insetvatn | 1914-1958 | 43 | 152 (133-181) | 45 | 296 (279-322) | 45 | 314 (291-367) | 43 | 145 (109-181) |
| 87 | Oksfjordvatn | 1955-1970 | 15 | 156 (135-170) | 5 | 339 (301-329) | 15 | 339 (319-352) | 5 | 171 (140-301) |
| 88 | Lille Mattisvatn | 1950-1972 | 16 | 139 (129-151) | 11 | 298 (280-314) | 16 | 319 (296-246) | 11 | 168 (143-312) |
| 89 | Lille Ropelvvann | 1957-1984 | 27 | 161 (128-172) | 11 | 309 (299-322) | 27 | 311 (294-332) | 11 | 152 (141-181) |
| 90 | Bjørnvatn | 1912-1967 | 55 | 151 (130-182) | 53 | 306 (286-327) | 55 | 311 (289-366) | 52 | 157 (117-189) |
| 91 | Murusjøen | 1926-2001 | 66 | 142 (121-155) | 74 | 327 (305-354) | 66 | 336 (311-366) | 65 | 184 (157-223) |
| 92 | Limingen | 1930-1960 | 27 | 152 (131-176) | 11 | 348 (289-385) | 30 | 369 (316-424) | 11 | 200 (120-240) |
| 93 | Vekteren | 1959-1986 | 23 | 157 (144-168) | 46 | 321 (296-341) | 15 | 339 (305-413) | 20 | 164 (145-190) |
| 94 | Saksvatn | 1960-1990 | 31 | 164 (141-180) | 29 | 306 (279-331) | 28 | 309 (298-334) | 29 | 147 (174-119) |
| 95 | Lenglingen | 1925-2003 | 76 | 144 (118-158) | 76 | 329 (307-383) | 77 | 339 (312-385) | 74 | 187 (157-235) |
| 96 | Engeren | 1911-1983 | 72 | 139 (119-157) | 72 | 347 (299-396) | 71 | 350 (311-386) | 71 | 204 (156-244) |
| 97 | Femunden | 1900-1995 | 82 | 148 (128-173) | 83 | 328 (305-353) | 83 | 343 (313-386) | 79 | 177 (152-214) |
| 98 | Isteren | 1950-1986 | 34 | 148 (113-157) | 35 | 309 (283-335) | 35 | 319 (291-385) | 34 | 162 (134-206) |
| 99 | Møkeren | 1911-2007 | 65 | 121 (91-141) | 47 | 332 (261-363) | 65 | 341 (303-446) | 37 | 212 (128-244) |
| 100 | Søndre Øyersjøen | 1960-2005 | 42 | 126 (99-138) | 18 | 334 (305-363) | 39 | 336 (308-367) | 18 | 210 (179-240) |
| 101 | Varalden | 1950-1983 | 26 | 123 (91-135) | 22 | 335 (312-367) | 27 | 350 (315-378) | 19 | 210 (181-245 |
| 102 | Rømsjøen | 1945-1995 | 46 | 119 (33-138) | 44 | 339 (305-376) | 48 | 359 (333-398) | 43 | 224 (171-293) |







**Appendix 3**.
Variation in average time of ice break-up, time of lake freeze-up, time when lake is completely
frozen, and length of ice-free period. The full model is formulated as (see description of parameters
in the main text):
$Y = \mu + \alpha_1 Alt + \alpha_2 Lat + \alpha_3 Long + \alpha_4 Alt*Lat + \alpha_5 Alt*Long + \alpha_6 Long*Lat + \alpha_7 Alt*Long*Lat + \alpha_8 Distance$
$+ \alpha_9 Area + \alpha_{10} Catch + \alpha_{11} Flow + \varepsilon$
Selection of the best model was based on AIC. The full model and the three best models are
presented, with the best model given in bold. AIC and $\Delta$AIC is given.
**Time of ice break-up:**

| No. | Model formulation (n = 101) | AIC | ΔAIC |
|---|---|---|---|
| 0 | Full model | 695.8 | 5.0 |
| **1** | **$Y = \mu + \alpha_1 Alt + \alpha_2 Lat + \alpha_3 Long + \alpha_4 Alt*Lat + \alpha_5 Alt*Long + \alpha_6 Long*Lat + \alpha_7 Alt*Long*Lat + \alpha_{11} Flow$** | **690.8** | **0** |
| 2 | $Y = \mu + \alpha_1 Alt + \alpha_2 Lat + \alpha_3 Long + \alpha_4 Alt*Lat + \alpha_5 Alt*Long + \alpha_6 Long*Lat + \alpha_7 Alt*Long*Lat + \alpha_8 Distance + \alpha_{11} Flow$ | 691.2 | 0.4 |
| 5 | $Y = \mu + \alpha_1 Alt + \alpha_2 Lat + \alpha_3 Long + \alpha_4 Alt*Lat + \alpha_5 Alt*Long + \alpha_6 Long*Lat + \alpha_7 Alt*Long*Lat$ | 691.7 | 0.9 |


**Time of lake freeze-up:**

| No. | Model formulation (n = 86) | AIC | ΔAIC |
|---|---|---|---|
| 0 | Full model | 719.5 | 11.8 |
| **1** | **$Y = \mu + \alpha_1 Alt + \alpha_2 Lat + \alpha_3 Long + \alpha_4 Alt*Lat + \alpha_6 Long*Lat + \alpha_8 Area$** | **707.7** | **0** |
| 2 | $Y = \mu + \alpha_1 Alt + \alpha_2 Lat + \alpha_3 Long + \alpha_6 Long*Lat + \alpha_8 Area$ | 708.1 | 0.4 |
| 3 | $Y = \mu + \alpha_1 Alt + \alpha_2 Lat + \alpha_3 Long + \alpha_4 Alt*Lat + \alpha_5 Alt*Long + \alpha_6 Long*Lat + \alpha_8 Area$ | 709.6 | 1.9 |


**Time when lake is completely frozen:**

| No. | Model formulation (n = 97) | AIC | ΔAIC |
|---|---|---|---|
| 0 | Full model | 838.0 | 12.5 |
| **1** | **$Y = \mu + \alpha_1 Alt + \alpha_2 Lat + \alpha_3 Long + \alpha_6 Long*Lat + \alpha_8 Area$** | **825.5** | **0.0** |
| 2 | $Y = \mu + \alpha_1 Alt + \alpha_2 Lat + \alpha_3 Long + \alpha_8 Area$ | 827.6 | 2.1 |



| 3 | $Y = \mu + \alpha_1 Alt + \alpha_2 Lat + \alpha_3 Long + \alpha_5 Alt*Long + \alpha_8 Area$ | 828.3 | 2.8 |


**Length of ice-free period:**

| No. | Model formulation (n = 86) | AIC | ΔAIC |
|-----|---------------------------|-----|------|
| 0 | Full model | 808.4 | 13.8 |
| **1** | **$Y = \mu + \alpha_1 Alt + \alpha_2 Lat + \alpha_3 Long + \alpha_5 Alt*Long + \alpha_6 Long*Lat + \alpha_8 Area$** | **794.6** | **0** |
| 2 | $Y = \mu + \alpha_1 Alt + \alpha_2 Lat + \alpha_3 Long + \alpha_4 Alt*Lat + \alpha_6 Long*Lat + \alpha_8 Area + \alpha_{10} Flow$ | 795.2 | 0.6 |
| 3 | $Y = \mu + \alpha_1 Alt + \alpha_2 Lat + \alpha_3 Long + \alpha_4 Alt*Lat + \alpha_6 Long*Lat + \alpha_8 Area + \alpha_9 Catch$ | 796.2 | 1.6 |






**Appendix 4**.
**Test for temporal variation in time of lake freeze-up, time when lake is completely frozen, and**
**length of ice-free period for 98 lakes in Norway**. Lake identity is modelled as a random factor, and
year is always included in the model as a fixed effect. The full model is formulated as (see description
of parameters in the main text):
$Y = \mu + \alpha_1 Alt + \alpha_2 Lat + \alpha_3 Long + \alpha_4 Alt*Lat + \alpha_5 Alt*Long + \alpha_6 Long*Lat + \alpha_7 Alt*Long*Lat + \alpha_8 Distance +$
$\alpha_9 Area + \alpha_{10} Catch + \alpha_{11} Flow + \alpha_{12} Year + \alpha_{13} Impounded + \alpha_{14} Amplitude + \varepsilon.$
Selection of the best model was based on AIC. The full model and the three best models are
presented, with the best model given in bold. AIC and ΔAIC is given.
**Time of lake freeze-up:**

| No. | Model formulation | AIC | ΔAIC |
|---|---|---|---|
| 0 | Full model | 25 776.8 | 63.7 |
| **1** | $Y = \mu + \alpha_1 Alt + \alpha_2 Lat + \alpha_3 Long + \alpha_8 Area + \alpha_{12} Year + \alpha_{13} Impounded + \alpha_{14} Amplitude$ | **25 713.1** | **0** |
| 2 | $Y = \mu + \alpha_1 Alt + \alpha_2 Lat + \alpha_3 Long + \alpha_6 Long*Lat + \alpha_9 Area + \alpha_{12} Year + \alpha_{13} Impounded + \alpha_{14} Amplitude$ | 25 713.8 | 0.7 |
| 3 | $Y = \mu + \alpha_1 Alt + \alpha_3 Long + \alpha_9 Area + \alpha_{12} Year + \alpha_{13} Impounded + \alpha_{14} Amplitude$ | 25 716.6 | 3.5 |


**Time when lake is completely frozen:**

| No. | Model formulation | AIC | ΔAIC |
|---|---|---|---|
| 0 | Full model | 35 781.8 | 67.1 |
| **1** | $Y = \mu + \alpha_1 Alt + \alpha_2 Lat + \alpha_3 Long + \alpha_6 Long*Lat + \alpha_9 Area + \alpha_{12} Year + \alpha_{13} Impounded + \alpha_{14} Amplitude$ | **35 714.7** | **0** |
| 2 | $Y = \mu + \alpha_1 Alt + \alpha_2 Lat + \alpha_3 Long + \alpha_8 Area + \alpha_{11} Year + \alpha_{12} Impounded + \alpha_{13} Amplitude$ | 35 715.0 | 0.3 |
| 3 | $Y = \mu + \alpha_1 Alt + \alpha_2 Lat + \alpha_3 Long + \alpha_6 Long*Lat + \alpha_8 Area + \alpha_{11} Year$ | 35 716.6 | 1.9 |


**Length of ice-free period:**

| No. | Model formulation | AIC | ΔAIC |
|---|---|---|---|





| 0 | Full model | 27 547.9 | 55.9 |
|---|---|---|---|
| **1** | **Y = μ + $\alpha_1$Alt + $\alpha_2$Lat + $\alpha_3$Long + $\alpha_6$Long\*Lat + $\alpha_9$Area+ $\alpha_{12}$Year + $\alpha_{13}$Impounded + $\alpha_{14}$Amplitude** | **27 492.0** | **0** |
| 2 | Y = μ + $\alpha_1$Alt + $\alpha_2$Lat + $\alpha_3$Long + $\alpha_8$Area + $\alpha_{11}$Year + $\alpha_{12}$Impounded + $\alpha_{13}$Amplitude | 27 494.1 | 2.1 |
| 3 | Y = μ + $\alpha_1$Alt + $\alpha_2$Lat + $\alpha_3$Long + $\alpha_6$Long\*Lat + $\alpha_8$Area + $\alpha_{11}$Year | 27 496.7 | 4.7 |






**Appendix 5**.
**Test for temporal variation in time of ice break-up.** Lake identity is modelled as a random factor,
and year is always included in the model as a fixed effect. NAO is included in the model as both a
linear and a non-linear effect. The full model is formulated as (see description of parameters in the
main text):
$Y = \mu + \alpha_1 Alt + \alpha_2 Lat + \alpha_3 Long + \alpha_4 Alt*Lat + \alpha_5 Alt*Long + \alpha_6 Long*Lat + \alpha_7 Alt*Long*Lat + \alpha_8 Distance +$
$\alpha_9 Area + \alpha_{10} Catch + \alpha_{11} Flow + \alpha_{12} Year + \alpha_{13} Impounded + \alpha_{14} Amplitude + \alpha_{15} NAO + \alpha_{16} NAO^2 + \varepsilon.$
Selection of the best model was based on AIC. The full model and the three best models are
presented, with the best model given in bold. AIC and ΔAIC is given.

| No. | Model formulation | AIC | ΔAIC |
|-----|-------------------|-----|------|
| 0 | Full model | 33 367.0 | 56.0 |
| 1 | $Y = \mu + \alpha_1 Alt + \alpha_2 Lat + \alpha_3 Long + \alpha_5 Alt*Long + \alpha_6 Long*Lat + \alpha_{12} Year + \alpha_{15} NAO$ | 33 311.0 | **0** |
| 2 | $Y = \mu + \alpha_1 Alt + \alpha_2 Lat + \alpha_3 Long + \alpha_6 Long*Lat + \alpha_{12} Year + \alpha_{15} NAO$ | 33 311.2 | 0.2 |
| 3 | $Y = \mu + \alpha_1 Alt + \alpha_2 Lat + \alpha_3 Long + \alpha_5 Alt*Long + \alpha_{12} Year + \alpha_{15} NAO$ | 33 315.5 | 4.5 |





**Appendix 6.**
**Test for non-linear temporal trends in ice phenology in 30-years periods.** Lakes with >50 years of
records of both date of break-up and date of frozen lake.

| Lake no | Lake | Period | Break up n | Break up Median | Freeze up n | Freeze up Median | Frozen lake n | Frozen lake Median | Ice free period n | Ice free period Median |
|---|---|---|---|---|---|---|---|---|---|---|
| 1 | Mjøsa (Hamar) | 1910-2001 | 76 | 111 (23-139) | 74 | 383 (318-440) | 63 | 392 (350-435) | 63 | 272 (208-401) |
| 2 | Storsjø | 1910-2011 | 66 | 124 (97-140) | 48 | 361 (333-392) | 76 | 390 (349-443) | 28 | 239 (200-276) |
| 3 | Lomnessjøen | 1919-1997 | 66 | 131 (96-147) | 69 | 320 (281-352) | 54 | 327 (302-379) | 58 | 186 (152-248) |
| 5 | Olstappen | 1967-2020 | 53 | 142 (129-158) | | | 52 | 309 (285-329) | | |
| 6 | Aursunden | 1902-2020 | 115 | 152 (129-175) | 58 | 314 (295-332) | 116 | 324 (295-355) | 57 | 158 (127-186) |
| 7 | Atnsjøen | 1917-2020 | 87 | 145 (122-165) | 95 | 320 (302-347) | 98 | 328 (312-363) | 84 | 176 (144-213) |
| 11 | Tesse | 1908-2020 | 74 | 148 (121-167) | | | 76 | 330 (311-363) | | |
| 12 | Aursjø | 1967-2020 | 53 | 169 (148-181) | | | 53 | 310 (293-332) | | |
| 13 | Breidalsvatn | 1967-2020 | 53 | 168 (147-191) | | | 53 | 323 (303-347) | | |
| 14 | Raudalsvatn | 1967-2020 | 53 | 157 (136-176) | | | 53 | 329 (313-365) | | |
| 17 | Kaldfjorden | 1967-2020 | 53 | 159 (136-170) | | | 53 | 309 (285-332) | | |
| 19 | Vinstern | 1950-2020 | 64 | 163 (147-181) | | | 69 | 317 (288-339) | | |
| 21 | Bygdin | 1950-2020 | 64 | 170 (153-185) | 15 | 326 (301-382) | 65 | 370 (315-416) | 14 | 157 (130-221) |
| 26 | Volbufjorden | 1920-1974 | 55 | 137 (119-150) | 54 | 320 (305-344) | 55 | 324 (312-353) | 54 | 184 (164-214) |
| 27 | Øyangen | 1919-1984 | 65 | 149 (130-168) | 62 | 318 (299-343) | 62 | 321 (304-344) | 61 | 170 (137-200) |
| 31 | Bergsjø | 1953-2020 | 58 | 160 (146-175) | 47 | 304 (288-343) | 56 | 314 (294-350) | 47 | 144 (127-170) |
| 33 | Krøderen | 1900-1964 | 64 | 124 (100-161) | 7 | 335 (315-366) | 60 | 338 (306-372) | 7 | 214 (189-255) |
| 35 | Tunhovdfjorden | 1920-2020 | 73 | 142 (119-161) | 45 | 329 (275-353) | 77 | 335 (305-362) | 41 | 186 (142-219) |
| 39 | Hjartsjå | 1919-1998 | 74 | 121 (91-139) | 43 | 328 (311-354) | 70 | 334 (313-388) | 42 | 207 (184-261) |
| 44 | Sandvinvatn | 1908-1998 | 59 | 106 (33-131) | 61 | 383 (224-437) | 64 | 398 (359-453) | 46 | 276 (225-342) |
| 45 | Vangsvatn | 1898-1989 | 69 | 113 (38-138) | 46 | 347 (316-402) | 78 | 354 (327-420) | 61 | 236 (197-333) |
| 46 | Vassbygdvatn | 1915-1987 | 69 | 116 (56-139) | 56 | 356 (277-401) | 65 | 371 (330-435) | 54 | 242 (158-305) |
| 48 | Veitastrondvatn | 1918-1991 | 65 | 137 (76-152) | 52 | 353 (311-416) | 61 | 356 (326-428) | 50 | 217 (171-284) |
| 51 | Nautsundvatn | 1908-1983 | 55 | 106 (33-130) | 75 | 353 (314-426) | 75 | 353 (314-426) | 54 | 248 (215-348) |
| 52 | Hestadfjorden | 1914-1995 | 70 | 117 (17-140) | 75 | 358 (320-423) | 77 | 371 (323-446) | 65 | 242 (192-382) |
| 55 | Lovatn | 1899-1979 | 72 | 108 (18-132) | 44 | 388 (347-436) | 51 | 388 (355-440) | 42 | 281 (227-395) |
| 68 | Namsvatn | 1908-1968 | 57 | 163 (137-184) | 19 | 319 (301-341) | 58 | 323 (291-351) | 17 | 164 (126-183) |
| 71 | Tustervatn | 1907-1968 | 54 | 156 (137-178) | 44 | 328 (304-366) | 50 | 343 (308-391) | 41 | 174 (127-216) |
| 76 | Kobbvatn | 1916-1978 | 61 | 149 (128-167) | 58 | 330 (304-386) | 60 | 339 (310-392) | 56 | 185 (140-245) |
| 90 | Bjørnvatn | 1912-1967 | 55 | 151 (130-182) | 53 | 306 (286-327) | 55 | 311 (289-366) | 52 | 157 (117-189) |
| 91 | Murusjøen | 1926-2001 | 66 | 142 (121-155) | 74 | 327 (305-354) | 66 | 336 (311-366) | 65 | 184 (157-223) |
| 95 | Lenglingen | 1925-2003 | 76 | 144 (118-158) | 76 | 329 (307-383) | 77 | 339 (312-385) | 74 | 187 (157-235) |
| 96 | Engeren | 1911-1983 | 72 | 139 (119-157) | 72 | 347 (299-396) | 71 | 350 (311-386) | 71 | 204 (156-244) |
| 97 | Femunden | 1900-1995 | 82 | 148 (128-173) | 83 | 328 (305-353) | 83 | 343 (313-386) | 79 | 177 (152-214) |
| 99 | Møkeren | 1911-2007 | 65 | 121 (91-141) | 47 | 332 (261-363) | 65 | 341 (303-446) | 37 | 212 (128-244) |
