# Peer review of "Geographic variation and temporal trends in ice phenology in Norwegian lakes over a century 2 3 Jan Henning L'Abée-Lund1, Leif Asbjørn Vøllestad2, John Edward Brittain1,3, Ånund Sigurd Kvambekk1 4 and Tord Solvang1 5 1 Norwegian Wat"

_The Cryosphere, 2020_

## Author Comment (AC3)

Below please find a detailed point-by-point response to the review comments with changes in the revised manuscript highlighted. Please also find a marked-up manuscript version showing the changes made.
* * *
**Reply to RC1 for the manuscript "Geographic variation and temporal trends in ice phenology in Norwegian lakes over a century" by Jan Henning L'Abée-Lund, Leif Asbjørn Vøllestad, John Edward Brittain, Ånund Sigurd Kvambekk, and Tord Solvang.**

Dear Andrew Newton Referee #1, thank you for your constructive comments on our manuscript. Please find below our responses to all of the review comments (in *bold and italic*). The resulting manuscript changes are highlighted in red.

This paper provides new insights on the variability of lake and river ice phenology in Norway. In general, I like this paper, it represents an area of the Northern Hemisphere that has been historically limited to only a few studies on ice phenology due to data availability, for that alone it is a good contribution. The writing is generally clear and the discussion makes sense based on the presented results. I have made some suggestions in the line by line comments, but there are two main issues that I would like to see addressed before publication:

Firstly, whilst I am somewhat constrained with a limited experience on the modelling component, I do feel this could be better explained and integrated into the methods section rather than hidden in the appendix. Some text that more clearly defines this part of the workflow would be welcomed – at the moment it is not as easy as it could be if someone wanted to replicate the workflow.

Reply: We included a more detailed description of the statistical modelling in the material and methods chapter to make it clearer.

Secondly, a couple of the figures need to be developed a little further as they are currently lacking sufficient information to ensure that they cannot be misunderstood. I have added more details on this below, but it mainly relates to ensuring they are adequately annotated (axis labels etc.) and the captions are more detailed. I have also put in a suggestion for modifications to Figure 1, this is not a request, just a suggestion that I think would help the authors with some of the points the make on variation of the site parameters.

Reply: The figures have been checked for lacking information and corrected accordingly.

Title: Perhaps change to "...over the last century".

Reply: The title is adjusted according to Ref. #2.

40-41: State that you mean surface area.

Reply: corrected.

48-50: Need to clarify that freeze up and breakup dates have changed over time.

Reply: The text has been improved.

51-53: Needs a little more information on the studies that are being referenced -e.g. data coverage, timespans etc. That context is important as the sentence too simple.

Reply: The time period studied is included for each reference.

57: "...considerably since 1880..."

Reply: corrected

58-61: I am an author on this paper and the revised version is about to be submitted in the next couple of days. The general trend is the same of an increased magnitude of change, but the time period was changed to 1976-2005 due to reviewer feedback. Thus, you may need to reconsider this sentence if the Newton and Mullan (2020) is accepted for publication.

Reply: Time period studied has been corrected.

81-85: Sentences on variation in topography and climate might benefit from a little more detail/referencing on what the differences are.

Reply: A description of the topography is included.

90-92: This is quite an important justification and an additional sentence highlighting how detecting the different influences are important for projecting future change might be useful.

Reply: An additional sentence is included.

97-99: It could be very helpful for the reader to see on Figure 1 some more geographical data that help to highlight these different things. E.g. a different colour for sites in the climatic zones, perhaps also an underlying topography to provide some additional context, or colour code the elevation. One idea might be to use a ramp scale for elevation for the inside colour of the circle, and then three different colours for the outline to show which climatic zone the site is in. This would be a nice succinct way to show the reader the main geographic characteristics of the sites. It could even be simpler, maybe as a series of repeat panels where the colours change with relation to climatic zone, elevation, area, depth etc. Note, this is just a suggestion that I think would be quite useful and is worth thinking about.

Reply: Topographical information has been included in the revised Figure 1.

117-118: Do you have any idea on what the magnitude of any errors might be when estimating these dates?

Reply: A comment of the magnitude of error has been included.

122-123: Similar to above point, perhaps you can clarify how you decide the date, what percentage of surface area needs to have ice for it to be declared frozen?

Reply: The text has been extended to clarify the procedure in deciding the ice phenology dates.

170-176: This section needs some extra information and integration of the information from the appendix into the main text. At the moment this part of the methods is a little vague and I am not 100% sure I follow the exact workflow. Whilst I am not an expert on this type of method, it would help if some extra context on the method and how it was applied in this study was added, or as a minimum extra references that can direct the reader to relevant details.

Reply: We have included more text in this chapter to make it clearer.

193: I am not quite sure I understand what you mean by "the year effect". This section needs a little more information to improve meaning of the text.

Reply: We have included some information to improve the meaning of the text.

285: Change to "...date of break-up generally becomes later with increasing latitude".

Reply: Corrected as suggested.

285-286: Clarify what you mean by circulation, I know what you mean, but could potentially be confused for water circulation. Perhaps "...macro- to local-scale atmospheric circulation and lake characteristics...".

Reply: Corrected as suggested.

289: "We found..." – this sentence needs rewording. Perhaps "Ice breakup dates are shown to be 2.3 days later with each degree of higher latitude", or something like that, needs rewording.

Reply: Corrected as suggested.

292: I am not sure "discrepancy" is really the right way to describe it. We would expect different regions to vary in different ways – e.g. I am not sure there has to be a reason for the difference given that the study areas are so different, though you later hint at that, so perhaps think about how you frame that argument. Do not need to change anything, but just something to think about.

Reply: We have replaced discrepancy with difference.

302: Typo "one exception".

Reply: Corrected as suggested.

301-313: I think there needs to be a little bit of rewording here on how you refer to longitude. Longitude is not like latitude where there is a direct difference in insolation being received, by this I mean it is not really longitude in the strict sense, but more about distance from the coast. So in this case longitude is a proxy for something else, such as continentality. You urge the caution here, rightly, and it might be worth just making sure the language is clearer.

Reply: The text has been expanded.

318: Every degree of latitude higher or lower? I know what you mean, but clarify.

Reply: The text has been improved.

346: Reword to "...earlier break-up, later freeze-up and later completely frozen lakes..."

Reply: Corrected as suggested.

352-360: Just another note that the Newton and Mullan paper, the revised version is about to be submitted, so it might be worth checking against this when you revise the manuscript. The broad inferences you have made will still be relevant but the discussion on the zero degree isotherm is no longer in the paper due changes to the time periods studied. You can probably still make the assertion if you perhaps plotted the isotherms on your own map maybe? You can get the data from the reanalysis project data and reworking in GIS. I would be perfectly happy to forward on the shapefiles presented in that original paper if you cannot get access.

Reply: We have included a new Figure 6 showing the zero-degree isotherm for the last two periods (1961-1990 and 1991-2020) to show the change. We chose these two periods as the modelling show significant change between them. The text has been changes accordingly.

382: The biological consequences section is missing maybe a paragraph, or at least a couple of sentences, on what this might mean for the future. You have made models for the ice phenology dates, so it would be good to think, even just conceptually, how this might be used with future climate projections. Worthy of some discussion.

Reply: We have added more text to this paragraph and added some discussion on the potential effects of climate change.

Figure 1: Could benefit from some tidying up, such as the line of longitude that goes through the inset box. Perhaps also considering darkening the country outline. Might be helpful to colour-code the sites for whether they are in a boreal, subalpine or alpine setting.

Reply: This figure is presented in a new colour-code to show the topography.

Figure 3: I am struggling to get my head completely around how this figure is arranged. It needs more annotation/labelling and a much more detailed caption explaining what it is showing. Also, add in

**extra information on what the red line is and the light red shading into the caption, perhaps even add on a correlation coefficient for each, there is space for that. I think I can understand it, but it could be made easier for the reader.**

Reply: We have deleted this figure as the correlation coefficients are presented in Table 1. Instead, we have included a figure showing median date of freeze-up, frozen lake, and break-up. The elevation of each lake (<500 m a.s.l., 500-1000 m a.s.l., >1000 m a.s.l.) is shown by the colour.

**Figure 4: Similar to the comment on figure 3, this needs some greater annotation on what the lines and the shaded areas mean. Presumably these are lines of best fit? If so, could you perhaps put the correlation coefficients on as well?**

Reply: We have improved the quality by presenting three new panels. Hopefully the legend is also now clear.

**Reply to RC2 for the manuscript "Geographic variation and temporal trends in ice phenology in Norwegian lakes over a century" by Jan Henning L'Abée-Lund, Leif Asbjørn Vøllestad, John Edward Brittain, Ånund Sigurd Kvambekk, and Tord Solvang.**

Dear Anonymous Referee #2, thank you for your constructive comments on our manuscript. Please find below our responses to all of the review comments (in *bold and italic*). The resulting manuscript changes are highlighted in red.

This paper presents lake ice phenology data set (over 100 lakes) from Norway and investigates and discusses spatial and temporal changes in ice cover. According to knowledge of the referee, Norwegian lake ice data has not been presented or analysed in this extent earlier in a scientific peer-reviewed article.

In general, lake ice phenology in Northern Hemisphere (NH) and Northern Europe is quite much studied and study idea is not showing dramatical novelty using GLM models. There are several previous studies showing trends of ice cover in NH, and also linkage with the NAO and regression models have been used for a long time. The data presented in this study is a valuable new data set itself.

The paper is well and clearly written and easy to follow. Data and methods are clearly presented. The modelling part could be explained more in detail in methods (formulas). More scientific references could be added in the introduction and discussion to show and acknowledge previous studies in the theme. The figures should be improved, they look a little bit prepared in a hurry and the finalization for scientific paper is missing. As a general conclusion, I recommend accept this paper with minor revisions (detailed comments below) after comments have been taken into account.

Title: "over a century" could be substituted with specific years i.e. '1890-2020'

Reply: Title adjusted accordingly.

**Introduction: row 79: Should Master Thesis of Solvang (2013) to be mentioned here? http://urn.nb.no/URN:NBN:no-39915**

Reply: Included

**Some more references could be added, see in the end of my comments.**

Reply: We thank the referee for the list of references. We have included most of them in the introduction (lines 43, 58, 64, 71, 75, 90-100) where we found it appropriate.

**2.4. Title could include GLM methods and model could be explained more in details**

Reply: "Modelling" is included in the title, and the text has been improved and expanded.

**row 289-291. Could different rates of trends in latitude-wise be linked to elevation gradient of lakes?**

Reply: We have included a correlation analysis in the text to quantify the effect of longitude (line 248). This indicates that elevation has minor effect.

**Discussion: Some more references could be added**

Reply: We thank the referee for the list of references. We have included some of them in the discussion (line 378, 406, 409-412, 435, 438) where we found it appropriate.

**Conclusion is rather short and not giving numerical result overview. -> Could be extended little bit.**

Reply: We have extended the Conclusion paragraph and included some numerical results.

**Figure 1. Europe map is rather small and unclear, Figure quality (pixels) poor -> Enhance. Either make Europe scale bigger, or largen North European aspect adding Norway's neighbors on the map. Scale could be made for 50 km/100 km scale.**

Reply: Figure 1 is now improved and we have included information on the topography of Norway.

**Figure 2. Quality should be improved. Year to x-axis. Hard to really find exact information on a lake. Is there certain reason to put lakes in that order? Would make nicer picture starting numbering lakes in order of length of the records.**

Reply: Year is added to the x-axis and the mark for each 5-year is enhanced. The y-axis is widened. The lakes are numbered in a manner reflecting south-north-direction. We have considered reorganization of the data. This can probably be visually improved. However, the geographic position will disappear, so we have retained the original lay-out.

**Figure 3. Axis captions missing, not clear scales.**

Reply: We have deleted this figure as the correlation coefficients are presented in Table 1. Instead, we have included a figure showing median date of freeze-up, frozen lake, and break-up. The elevation of each lake (<500 m a.s.l., 500-1000 m a.s.l., >1000 m a.s.l.) is shown by the colour.

**Figure 4. There are not many points east from longitude 25 degrees. How would the figure look without them? Is the longitude correlated with elevation?**

Reply: Yes, the data set consists of few lakes east of 25 °E. We have included a correlation analysis in the text to quantify the effect of longitude (line 248).

**Figure 5. Is the starting period 1900, not 1901? (other decades start with year 1). Or is the one period 31 years? Please rewrite caption text, it is not clear.**

Reply: We have corrected the first 30-years period starting 1901. The figure is corrected after reanalysing the data.

**Appendix 1. Please check, and as defining decimals (in English) Check also km2 > km2 and m3 -> m3. Maybe framing the table would make easier to read. (Also to other similar tables)**

Reply: The table has been corrected according to English style, and framing the table and rows have been changed to make reading easier.

**Appendix 2. Would it be possible to show some of the median records on the map as different colors? Some kind of spatial visualization of results on the map would be good to add.**

Reply: We have presented a new map of the median records of the ice phenology variables, shown in Figure 3.

**Through the text: I would scan for word 'altitude' and assess if elevation would be better term.**

Reply: We have replaced altitude with elevation.

**Also spelling of break-up/break up and freeze-up should be used in uniform way.**

Reply: Corrected throughout manuscript.

**row 25. solar radian -> radiation?**

Reply: Corrected.

**row 105. Might be good to open nve.com NVE abbreviation "Norwegian Water Resources and Energy Directorate website"**

Reply: Full text now included.

**row 158. glm, usually used capital letters, GLM?**

Reply: We have changed to capitals.

**row 168. lake area -> lake surface area (Please check also through whole paper)**

Reply: Surface inserted in text.

row 178. freeze up -> freeze-up. Please check consistent way of writing break-up and freeze-up through the paper, including figures and tables.

Reply: Correction carried out.

**row 185. Impounded is not clear term to me? Regulated lake? (Check also Appendix 1)**

Reply: Impounded indicates a lake that has been regulated for hydropower. We use regulated as suggested.

row 276. Present day is maybe not good wording, if paper will be read also decades later ð maybe just write until year 2020.

Reply: Corrected.

row 283. conglomerate word sound odd to me, is it necessary to use this specific word?

Reply: Conglomerate is deleted.

row 299. remove extra comma

Reply: Corrected.

row 306. Baltic Ocean -> Baltic Sea

Reply: Corrected.

**row 329. There is more recent study by Korhonen (2006) showing isolines of freeze-up and break-up of Finnish lakes.**

Reply: Korhonen (2006) inserted in text and reference list.

row 544. warming world

Reply: Corrected.

row 579 and 583. missing space "period 1890-2020"

Reply: Corrected.

*row 614. Tabell 3 -> Table 3.*

Reply: Corrected.

row 615. CV could be opened up in table caption even though it is introduced in the article text

Reply: Corrected.

**References to be considered in introduction or discussion (in alphabetical order):**

Reply: We have found several of the references useful (marked with bold type) and have included them in introduction and/or discussion.

**Filazzola, A. Blagrave**, K. Imrit, MA. Sharma, S.: Climate change drives increases in extreme events for lake ice in the Northern Hemisphere, Geophysical Research Letters, e2020GL089608, 2020

**George, DG, Jarvinen**, M. & Arvola, L.: The influence of the North Atlantic Oscillation on the winter characteristics of Windermere (UK) and Paajarvi (Finland). Boreal Environ Res 9:389–399, 2004

**Korhonen**, J..: Long-term changes in lake ice cover in Finland, Nordic Hydrology 37(4-5), 347–363, 2006

Korhonen, J.: Long-term changes and variability of the winter and spring season hydrological regime in Finland, University of Helsinki, Doctoral dissertation, 2019.

Kuusisto, E.: An analysis of the longest ice observation series made on Finnish lakes, Aqua Fenn 17:123–132, 1987

**Kuusisto**, E. & Elo, A.-R.: Lake and river ice variables as climate indicators in Northern Europe, Verhandlungen des Internationalen Verein Limnologie 27(5), 2761–2764, 2000.

**Livingstone**, D.M.: Large-scale climatic forcing detected in historical observations of lake ice break-up, Verhandlungen der Internationalen Vereinigung für Limnologie 27, 2775–2783, 2000.

**Prowse, T.,** Alfredsen, K., Beltaos, S. et al.: Effects of Changes in Arctic Lake and River Ice, AMBIO 40, 63–74, 2011

**Sharma, S. &** Magnuson, J.J.: Oscillatory dynamics do not mask linear trends in the timing of ice breakup for Northern Hemisphere lakes from 1855 to 2004. Climatic Change 124(4), 835–847. 2014.

Šmejkalová, T., Edwards, M.E. & Dash, J.: Arctic lakes show strong decadal trend in earlier spring iceout, Scientific Reports 6, 38449. 2016.

Weyhenmeyer, G.A., Meili, M. & Livingstone, D.M.: Nonlinear temperature response of lake ice breakup, Geophysical Research Letters 31, L07203, 2004.

**Yoo, J. &** D'Odorico, P.: Trends and fluctuations in the dates of ice break-up of lakes in Northern Europe: the effect of the North Atlantic Oscillation, Journal of Hydrology 268(1), 100–112, 2002.